# Dynamin-related protein 1 deficiency accelerates lipopolysaccharide-induced acute liver injury and inflammation in mice

Lixiang Wang [1,2✉], Xin Li[3], Yuki Hanada [2], Nao Hasuzawa[4], Yoshinori Moriyama[4], Masatoshi Nomura [2,4✉] & Ken Yamamoto [1]

Mitochondrial fusion and fission, which are strongly related to normal mitochondrial function, are referred to as mitochondrial dynamics. Mitochondrial fusion defects in the liver cause a non-alcoholic steatohepatitis-like phenotype and liver cancer. However, whether mitochondrial fission defect directly impair liver function and stimulate liver disease progression, too, is unclear. Dynamin-related protein 1 (DRP1) is a key factor controlling mitochondrial fission. We hypothesized that DRP1 defects are a causal factor directly involved in liver disease development and stimulate liver disease progression. Drp1 defects directly promoted endoplasmic reticulum (ER) stress, hepatocyte death, and subsequently induced infiltration of inflammatory macrophages. Drp1 deletion increased the expression of numerous genes involved in the immune response and DNA damage in Drp1LiKO mouse primary hepatocytes. We administered lipopolysaccharide (LPS) to liver-specific Drp1-knockout (Drp1LiKO) mice and observed an increased inflammatory cytokine expression in the liver and serum caused by exaggerated ER stress and enhanced inflammasome activation. This study indicates that Drp1 defect-induced mitochondrial dynamics dysfunction directly regulates the fate and function of hepatocytes and enhances LPS-induced acute liver injury in vivo.

[1] Department of Medical Biochemistry, Kurume University School of Medicine, Kurume, Japan. [2] Department of Medicine and Bioregulatory Science, Graduate School of Medical Science, Kyushu University, Higashi-ku, Fukuoka, Japan. [3] Department of Molecular Pathobiology, New York University College of Dentistry, New York, NY, USA. [4] Division of Endocrinology and Metabolism, Department of Internal Medicine, Kurume University School of Medicine, Kurume, Japan. ✉email: ourika0211@med.kurume-u.ac.jp; nomura@med.kurume-u.ac.jp

Mitochondria play a critical role in maintaining hepatocyte integrity and function, and mitochondrial dysfunction leads to liver diseases[1–4]. Mitochondrial function and morphology are interdependent, and the latter is shaped by ongoing mitochondrial fusion and fission (mitochondrial dynamics)[5–7]. In vertebrates, mitofusin-1/2 (MFN1/2) and optic atrophy 1 (OPA1) control mitochondrial fusion, while dynamin-related protein 1 (DRP1) and its receptors control mitochondrial fission. Mitochondrial dynamics research in recent years has shown the functional importance of mitochondrial dynamics in liver diseases. Kim et al. (2013, 2014) reported the impact of viral infection on mitochondrial dynamics, and mitochondrial dynamics alterations are used by hepatitis B virus[8] and hepatitis C virus[9] for maintenance of persistent infection. Overall, mitochondrial dynamics disruption promotes viral pathogenesis. Mitochondrial dynamics and liver diseases are at a crossroads. Studies have reported abnormal mitochondrial dynamics in other liver pathophysiological conditions, too. Cadmium is a long-lived environmental and occupational pollutant, and cadmium hepatotoxicity induces DRP1-dependent mitochondrial fragmentation by disturbing calcium homeostasis[10]. Mitochondrial dynamics are also related to the mechanism underlying acetaminophen-induced acute liver damage. Acetaminophen changes mitochondrial DRP1 levels, and when the DRP1 inhibitor Mdivi-1 inhibits mitochondria fission, acetaminophen induces greater hepatic impairment[11]. Alcoholic animal models also show mitochondrial morphology alteration. Mitochondria in normal hepatocytes usually show relatively slow dynamics, which are sensitive to inhibition by ethanol exposure[12]. Mfn2 ablation in the liver causes endoplasmic reticulum (ER)-mitochondrial phosphatidylserine transfer defects, leading to a non-alcoholic steatohepatitis (NASH)-like phenotype and liver cancer[13].

We reported that a mitochondrial fission defect in liver-specific Drp1-knockout (Drp1LiKO) mice, which demonstrate ER stress-promoted fibroblast growth factor 21 expression, subsequently functions as a metabolic regulator with anti-obesity and anti-diabetic effects. In addition, hematoxylin and eosin (H&E) staining of Drp1LiKO mouse liver sections shows a disorganized lobular parenchyma with inflammatory cell infiltration[14]. Our previous studies have also reported that Drp1 disruption in the liver changes the expression of genes involved in the immune system in the liver. Gene Ontology (GO) biological studies have reported that 7 of the top 10 clusters are related to the immune system. These clusters include terms such as "immune response," "phagocytosis," "antigen processing," and "presentation"[15].

In this study, we hypothesized that Drp1 deficiency-induced mitochondrial fission defects directly lead to liver disease development. In addition, we hypothesized that Drp1 defects stimulate liver disease progression. The aim was to determine the relationship between mitochondrial fission defects and liver disease progression. We tested our hypotheses in an acute liver injury experimental mouse model via lipopolysaccharide (LPS)-induced endotoxin shock. Our study indicated that Drp1 defect-induced liver inflammation progression, which will provide insight into the role of Drp1 in liver function and acute liver injury.

## Results

### Increased inflammatory response and cell death in LPS-treated Drp1LiKO mice.
In control mice, LPS administration increased messenger RNA (mRNA) levels of tumor necrosis factor alpha (Tnfa), interleukin 6 (Il6), interleukin 1 beta (Il1b), interferon beta 1 (Ifnb1), monocyte chemoattractant protein-1 (Mcp1), and the NLR family pyrin domain containing 3 (Nlrp3), which peaked at 1 h in the liver (Supplementary Fig. 1a). LPS-induced liver TNF, IL-6, interleukin 10 (IL-10) and MCP1 levels peaked at 1 h, IL-1β levels peaked at 4 h, and interferon gamma (IFN-γ) levels peaked at 8 h (Supplementary Fig. 1b). LPS-induced serum TNF and IL-10 levels peaked at 1 h; MCP1 level peaked at 4 h; and IL-6 and IFN-γ levels peaked at 8 h (Supplementary Fig. 1c).

Next, we compared the responses to LPS in control and Drp1LiKO mice after LPS injection. Drp1LiKO mice developed more severe symptoms of endotoxin shock, such as lack of activity and hunched back posture. Consistent with this observation, we also observed a significant increase in serum TNF levels at 1 h and IL-6, IL-1β, MCP1, and IFN-γ levels at 8 h after LPS injection in Drp1LiKO mice compared to control mice (Fig. 1a). In addition, liver Tnfa, Il6, Il1b, Ifnb1, and Mcp1 mRNA expression levels also increased in LPS-treated Drp1LiKO mice compared to control mice. An interesting finding was a decrease in Drp1 mRNA levels in control mice, probably through transcriptional inhibition (Fig. 1b). We also detected a significant increase in liver TNF, IL-10, IL-1β, MCP1, and IFN-γ levels in LPS-treated Drp1LiKO mice (Fig. 1c).

To evaluate the degree of functional damage in the liver, we further analyzed the serum levels of alanine aminotransferase (ALT) and aspartate transaminase (AST), which are two markers of hepatocellular injure or necrosis. Both ALT and AST increased in LPS-treated Drp1LiKO mice (Fig. 1d), indicating increase of hepatocyte death in the absence of Drp1. Caspases are both initiators and effectors of apoptotic cascades[16]; we used western blot analysis to quantify the procaspase-3 and cleaved caspase-3 protein levels. We found that Drp1 deletion increased cleaved caspase-3 levels in both non-treated and LPS-treated Drp1LiKO mice compared to control mice (Fig. 2a and b).

### Increased ER stress response in LPS-treated Drp1LiKO mice.
Western blot analysis of the liver protein samples showed that LPS induces an inflammatory response via nuclear factor kappa B (NF-κB) and mitogen-activated protein kinase (MAPK) pathways, while there was no difference in NF-κB and MAPK pathways between control and Drp1LiKO mice (Supplementary Fig. 2a and b). In addition to these pathways, recent studies have reported that LPS promotes DRP1-dependent mitochondrial fission and associated inflammatory responses in different cell types[17,18]. In the present study, LPS decreased Drp1 mRNA levels in the liver (Fig. 1b); therefore, we examined LPS-regulated translational and post-translational modifications on DRP1. Phosphorylation of S616 (S635 in mice) and dephosphorylation of S637 (S656 in mice) are associated with the activation of DRP1 as well as its translocation to the mitochondria[19–21]. First, we found that LPS induced a continuous decrease in Drp1 protein levels in control mice (Fig. 2a and c); in contrast, LPS induced the robust phosphorylation of S635 at 1 h, indicating a transient increase in mitochondrial fission (Fig. 2a and d). Drp1 phosphorylation levels of S656 were not detected.

It has previously been demonstrated that the ER stress pathway is involved in inflammatory responses and the pathogenesis of various chronic inflammatory diseases[22–24]. In our previous study, we reported that Drp1 deletion in the liver caused ER stress via the eukaryotic translation factor 2α (P-eIF2α) pathway[14]. Phosphorylation of eIF2α levels increased in LPS-treated Drp1LiKO mice, indicating a further increase in ER stress in LPS-treated Drp1LiKO mice (Fig. 2a and e). ER stress evokes upregulation of activating transcription factor 3 (Atf3), DNA damage–inducible transcript 3 (Ddit3; Chop), nuclear protein 1 (Nupr1; p8), and Tribbles homolog 3 (Trib3), and these genes are related to ER stress and ER stress-mediated cell apoptosis[25–27]. In the current study, real-time polymerase chain reaction (PCR) analysis confirmed that the

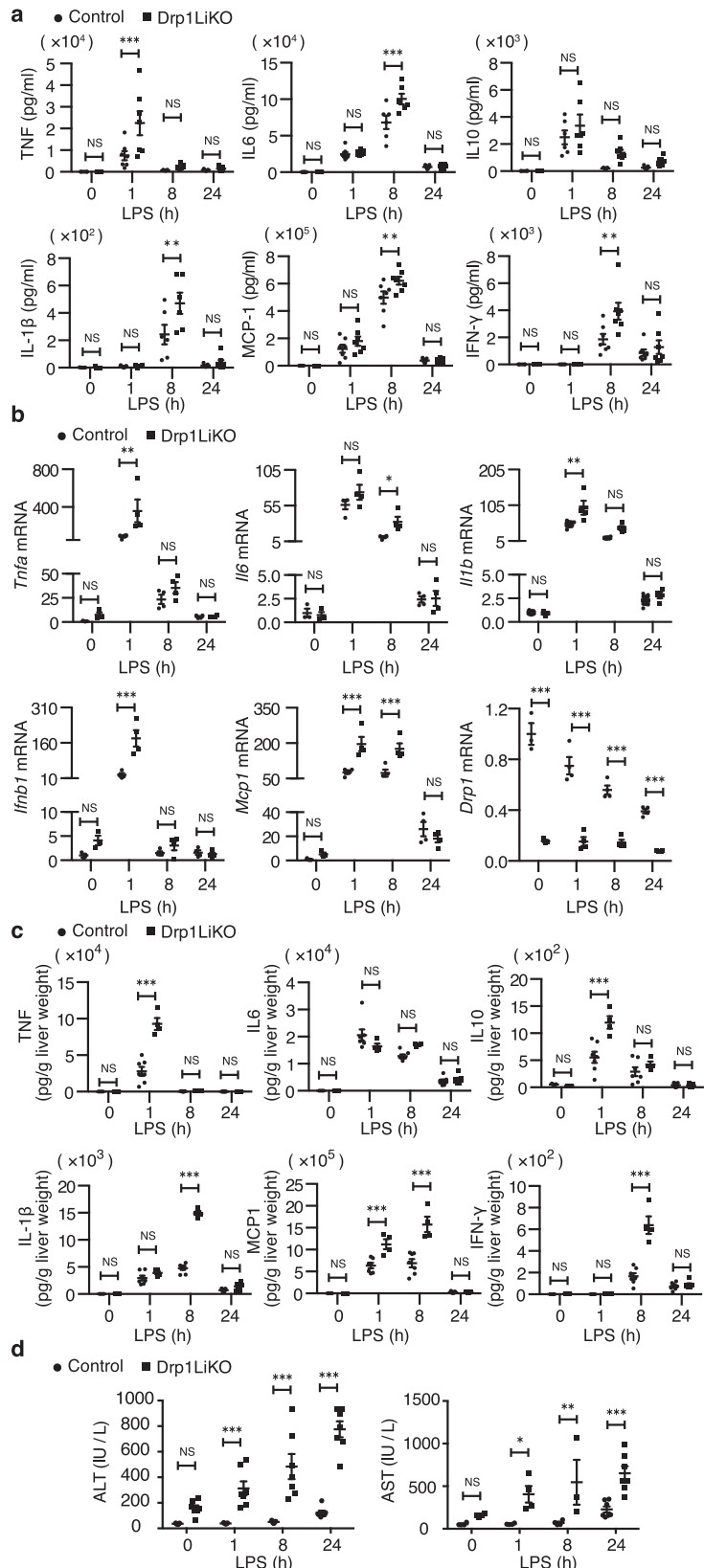

mRNA expression of these ER stress-related genes increased in LPS-treated *Drp1*LiKO mice livers (Fig. 2f).

**Inflammasome overactivation in LPS-treated *Drp1*LiKO mice**. Compared with control mice, western blot analysis showed that

LPS-treated *Drp1*LiKO mice increased the activation of NLRP3 inflammasome pathways during the LPS-induced inflammatory response in the liver as well as increased Nlrp3 inflammasome marker IL-1β and Nlrp3 levels (Fig. 2g and h). Gasdermin D, a newly discovered intracellular protein, forming pores regulate the secretion of cytokines such as IL-1β, in response to LPS and other

**Fig. 1 LPS-induced inflammatory response was augmented in *Drp1*LiKO mice.** Control and *Drp1*LiKO mice were treated with saline alone (referred to as LPS 0 h) or LPS (5 mg/kg, intraperitoneally). At indicated time points after LPS treatment, serum and liver samples were collected. **a** Serum TNF, IL-6, IL-10, IL-1β, MCP1, and IFN-γ levels were determined by BD cytometric bead array. Values are expressed as mean ± SEM. n = 5-7. **p < 0.01, ***p < 0.001 determined by Two-way ANOVA with Bonferroni's Post Hoc Test. **b** Liver *Tnfa*, *Il6*, *Il1b*, *Ifnb1*, *Mcp1*, and *Drp1* mRNA expression was determined by quantitative real-time PCR. Results were normalized to *Gapdh* expression and are shown as fold-changes relative to gene expression in saline-treated control mice. Values are expressed as mean ± SEM. n = 3-4 *p < 0.05, **p < 0.01, ***p < 0.001 determined by Two-way ANOVA with Bonferroni's Post Hoc Test. **c** Liver TNF, IL-6, IL-10, IL-1β, MCP1 and IFN-γ levels were determined by the BD cytometric bead array. Values are expressed as mean ± SEM. n = 3-7. ***P < 0.001 determined by Two-way ANOVA with Bonferroni's *P*ost Hoc Test. **d** Serum ALT and AST levels were measured using the DRI-CHEM3500 Chemistry Analyzer. n = 4-7 *p < 0.05, **p < 0.01, ***p < 0.001 determined by Two-way ANOVA with Bonferroni's Post Hoc Test. LPS lipopolysaccharide; *Drp1*LiKO, liver-specific *Drp1*-knockout, TNF tumor necrosis factor, IL interleukin, MCP1 monocyte chemoattractant protein-1, IFN-γ interferon gamma, PCR polymerase chain reaction, *Gapdh* glyceraldehyde 3-phosphate dehydrogenase, ALT alanine aminotransferase, AST aspartate transaminase, SEM standard error of the mean, NS no significant difference.

activators of the inflammasome[28–30]. We also observed a significant increase in serum gasdermin D levels at 8 h and 24 h after LPS injection in *Drp1*LiKO mice compared to control mice (Fig. 2i). Macrophages represent a key cellular component of the liver and are essential for maintaining tissue homeostasis and ensuring rapid response to hepatic injure[31,32]. Given that activation of the inflammasome pathway increased in *Drp1*LiKO mice, we investigated inflammasome activation in macrophages. To induce the maximum release of IL-1β, mice received intraperitoneal injection of 10-mg/kg body weight LPS for 4 h and then an additional 50 μl of 100-mM adenosine triphosphate (ATP) 30 min before sacrifice. H&E staining confirmed inflammatory cell infiltration in both saline- and LPS/ATP-treated livers of *Drp1*LiKO mice (Fig. 3a). Terminal deoxynucleotidyl transferase dUTP nick end labeling (TUNEL) staining showed that apoptotic hepatocytes increased in saline-treated (Ctrl 8.67 ± 1.86 vs. KO 47.33 ± 6.17) and LPS/ATP-treated (Ctrl 42.00 ± 6.08 vs. KO 113.33 ± 10.87) livers of *Drp1*LiKO mice (Fig. 3a and b). Consistent with this observation, serum ALT and AST levels also increased in LPS/ATP-treated *Drp1*LiKO mice (Supplementary Fig. 3a). Consistent with our previous observations (Fig. 1b), liver *Tnfa*, *Il6*, *Il1b*, *F4/80*, *Mcp1*, and *Nlrp3* mRNA expression significantly increased in *Drp1*LiKO mice compared to control mice after LPS/ATP treatment (Supplementary Fig. 3b).

Macrophages, neutrophils, T-cells, and B-cells are cell types known to play roles in liver inflammation; therefore, to determine the composition of inflammatory cell clusters in *Drp1*LiKO mice and especially the cell types that are predominantly expressed in these cell clusters, we performed immunostaining in liver tissues with various immune markers, including adhesion G protein-coupled receptor E1 (F4/80) for macrophages, lymphocyte antigen 6 complex, locus G (Ly6G) for neutrophils, CD3 antigen, epsilon polypeptide (CD3) for T-cells, and protein tyrosine phosphatase, receptor type, C (B220) for B-cells (Fig. 3a and Supplementary Fig. 4a). Cell-counting results are summarized in Fig. 3c and Supplementary Fig. 4b-d. We found no differences in the number of T-cells (Supplementary Fig. 4b) and neutrophils (Supplementary Fig. 4d), but the number of B-cells increased (saline Ctrl 183.33 ± 26.64 vs. saline KO 367.00 ± 22.50; LPS/ATP Ctrl 206.33 ± 39.30 vs. LPS/ATP KO 382.33 ± 29.85) (Supplementary Fig. 4c), and F4/80-positive macrophages were the largest population of the increased inflammatory cells (saline Ctrl 2401.50 ± 243.40 vs. saline KO 6097.75 ± 1089.91; LPS/ATP Ctrl 4851.25 ± 167.44 vs. LPS/ATP KO 8630.50 ± 904.51) (Fig. 3c). Interestingly, immunostaining analysis also revealed co-immunostaining of IL-1β with F4/80-positive macrophages, indicating specific co-localization of IL-1β-positive cells with macrophages (Fig. 3d and Supplementary Fig. 4e). We observed a clear line of liver-resident Kupffer cells along the sinusoid wall, which constituted the main macrophage population in the livers of control mice (Fig. 3d, dashed lines, Magnified1, 2, 5, and 6). In

contrast, macrophages in *Drp1*LiKO mice, showed an abnormal morphology with irregular location and distribution patterns, some macrophages with low F4/80 expression showed IL-1β production even without LPS/ATP stimulation in *Drp1*LiKO mice but not in control mice (Fig. 3d, Magnified3 and Magnified4). In LPS/ATP treated *Drp1*LiKO mice, the majority of macrophages showed low F4/80 expression but high IL-1β expression after LPS/ATP treatment (Fig. 3d, Magnified7), and only a few macrophages showed high F4/80 expression but no IL-1β expression after LPS/ATP treatment (Fig. 3d, Magnified8).

It is possible that different populations of hepatic macrophages in control and *Drp1*LiKO mice exert distinct functions and contribute to differences in response to an acute LPS challenge. During acute inflammation, M1-polarized macrophages express high IL-1β levels, while IL-1β levels are undetectable in M2-polarized macrophages[33]. To explore the polarization signature of these macrophages in the liver of control and *Drp1*LiKO mice, we performed a flow cytometry assay and sequential gating analysis using isolated non-parenchymal liver cells without LPS injection. Macrophages were identified as 7AAD−CD45+CD11b+F4/80+ cells (Fig. 3e), and the expression of the pro-inflammatory macrophage (M1) markers CD64 and CD80 as well as anti-inflammatory macrophage (M2) marker CD206 was used to verify polarization. Sequential gating analysis showed a significant increase in CD80^HighCD64^Lo, CD80^LoCD64^High and CD80^HighCD64^High pro-inflammatory macrophages (Fig. 3f) but a decrease in CD206^High anti-inflammatory macrophages in *Drp1*LiKO mice (Fig. 3g). Therefore, Drp1 loss in hepatocytes induces macrophage activation, and most of these activated macrophages are prone to differentiating into pro-inflammatory macrophages.

**Decreased autophagy formation in LPS-treated *Drp1*LiKO mice.** Autophagy is reported to be involved in mitochondrial dynamics and mitochondrial quality control; furthermore, autophagosomes are known to form at the mitochondria-ER contact site (MAM) in mammalian cells[34]. In our previous study, we demonstrated that loss of Drp1 leads to MAM defects; therefore, we investigated whether Drp1 defects alter autophagy formation in LPS-treated hepatocytes. The autophagy pathways in the LPS-induced inflammatory response were activated, and the autophagy activation marker light chain 3-phospholipid-conjugated (LCLc3-II3-II) expression increased and peaked at 8 h after LPS treatment in control mice, while LCLc3-II3-II expression decreased in *Drp1*LiKO mice compared to control mice (Fig. 4a and b). It is known that the degradation of sequestosome 1 (SQSTM1/P62) by the autophagic-lysosome pathway and deficient autophagy can lead to P62 accumulation at the protein level[35,36]. In addition, western blot analysis showed an increase in P62 accumulation in *Drp1*LiKO mice (Fig. 4a and c). Selective autophagy of mitochondria, known as mitophagy, is an important

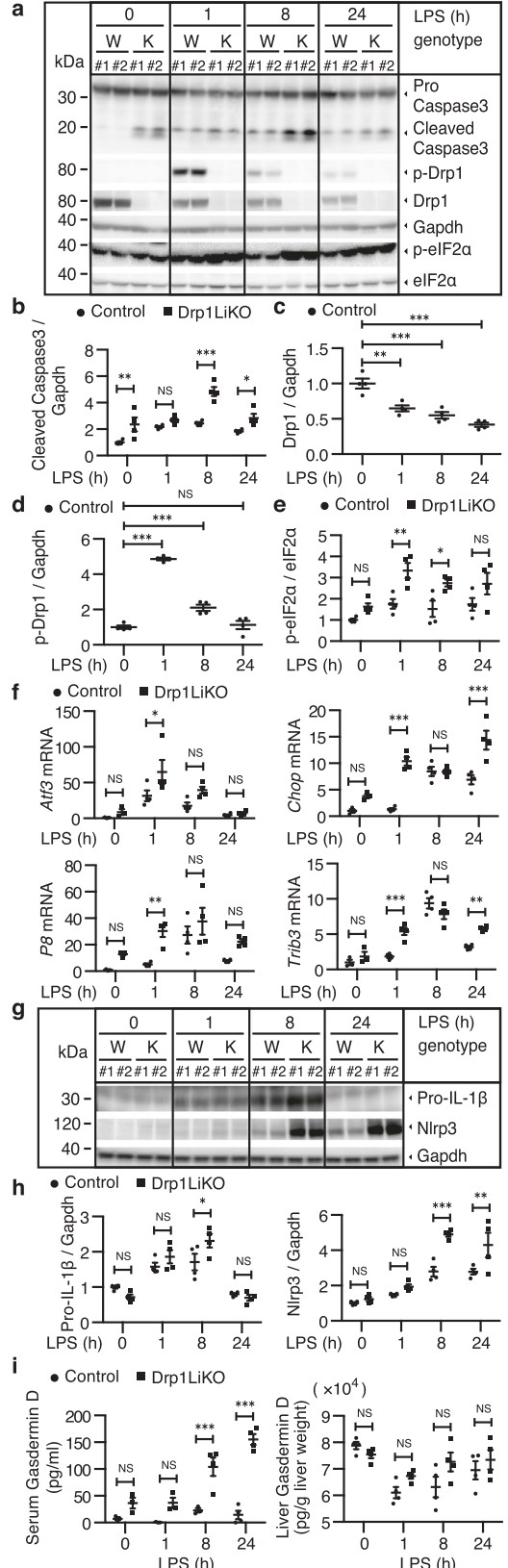

**Fig. 2 LPS-induced increased ER stress and NLRP3 inflammasome activation in the liver of *Drp1*LiKO mice.** Control and *Drp1*LiKO mice were treated with saline alone (referred to as LPS 0 h) or LPS (5 mg/kg, intraperitoneally). At indicated time points after LPS treatment, liver samples were collected. **a–e** Western blot analysis and densitometric quantification of caspase-3, p-Drp1(S635), Drp1, p-eIF2α, and eIF2α expression in saline- or LPS-treated liver lysates. Gapdh served as an internal control. The biological duplicate samples in each condition numbered as #1 and #2. Values are expressed as mean ± SEM ($n = 4$). $*p < 0.05$, $**p < 0.01$, and $***p < 0.001$ determined by two-way ANOVA with Bonferroni's post hoc test (**b** and **e**). $**p < 0.01$ and $***p < 0.001$ determined by one-way ANOVA with Dunnett's multiple comparisons test (**c** and **d**). **f** Liver *Atf3*, *Chop*, *P8*, and *Trib3* mRNA expression were determined by quantitative real-time PCR. Results are normalized to *Gapdh* expression and are shown as fold-changes relative to gene expression in saline-treated control mice. Values are expressed as mean ± SEM ($n = 3$–4). $*p < 0.05$, $**p < 0.01$, and $***p < 0.001$ determined by two-way ANOVA with Bonferroni's post hoc test. **g**, **h** Western blot analysis and densitometric quantification of Pro-IL-1β and Nlrp3 expression in saline- or LPS-treated liver lysates. Gapdh served as an internal control. The biological duplicate samples in each condition numbered as #1 and #2. Values are expressed as means ± SEM ($n = 4$). $*p < 0.05$, $**p < 0.01$, and $***p < 0.001$ determined by two-way ANOVA with Bonferroni's post hoc test. **i** Serum and liver gasdermin D levels were determined by Gasdermin D ELISA kit. Values are expressed as mean ± SEM. $n = 3$-4. $***p < 0.001$ determined by Two-way ANOVA with Bonferroni's Post Hoc Test. LPS lipopolysaccharide, *Drp1*LiKO liver-specific *Drp1*-knockout, eIF2α eukaryotic translation factor 2α, *Atf3* activating transcription factor 3, *Chop* DNA damage–inducible transcript 3 (*Ddit3*), *P8* nuclear protein 1 (*Nupr1*), *Trib3* Tribbles homolog 3, NLRP3 NLR family pyrin domain containing 3, IL interleukin, *Gapdh* glyceraldehyde 3-phosphate dehydrogenase, PCR polymerase chain reaction, SEM standard error of the mean, NS no significant difference.

*Drp1*LiKO mice. There was a greater increase in Pink1 levels in *Drp1*LiKO mice at 1 and 8 h, which was sustained to 24 h after LPS treatment (Fig. 4a and d), indicating that LPS treatment induces mitophagy impairment and that mitophagy impairment is further disturbed by Drp1 defects in *Drp1*LiKO mice.

Immunofluorescence microscopy showed a decrease in LC3Lc3-II-positive dots in *Drp1*LiKO mice compared to control mice after LPS/ATP treatment (Fig. 4e). Of note, localization of LC3-Lc3-positive cells was different from that of IL-1β-positive macrophages, indicating that autophagy occurs in hepatocytes but not in macrophages. Western blot analysis also confirmed that the LC3-ILc3-III in hepatocytes decreased in *Drp1*LiKO mice compared to control mice after LPS/ATP treatment (Fig. 4f).

**Increased cell death, inflammatory response, and decreased ΔΨ$_m$ in *Drp1*LiKO mouse primary hepatocytes.** So far, our results showed that Drp1 disruption in the liver can lead to mitophagy defects and hepatocyte apoptosis, which might trigger enhanced infiltration and polarization of macrophages. To gain further insight into this mechanism, we conducted additional experiments with primary hepatocytes from control and *Drp1* knockout mice. Isolation and culture of the hepatocytes were performed using a two-step collagenase perfusion method. When examined and counted under a light microscope using a hemocytometer chamber, the cell yield and viability were less in *Drp1*LiKO mice compared to control mice, indicating that *Drp1* disruption in the liver impairs hepatocyte survival (Fig. 5a).

Next, to fully reveal the gene expression altered by *Drp1* deletion, we conducted an independent microarray analysis with primary hepatocyte harvested 24 h after seeding. Genes were selected using the criterion of a Z score of ≥2, which identified

mitochondrial quality control mechanism[37]. Phosphatase and tensin homolog-induced putative kinase 1 (PINK1) is an established mediator of mitophagy[38–40]. In healthy mitochondria, which have a well-maintained mitochondrial membrane potential (ΔΨ$_m$), PINK1 levels remain low or undetectable. An acute LPS challenge led to Pink1 accumulation in both control and

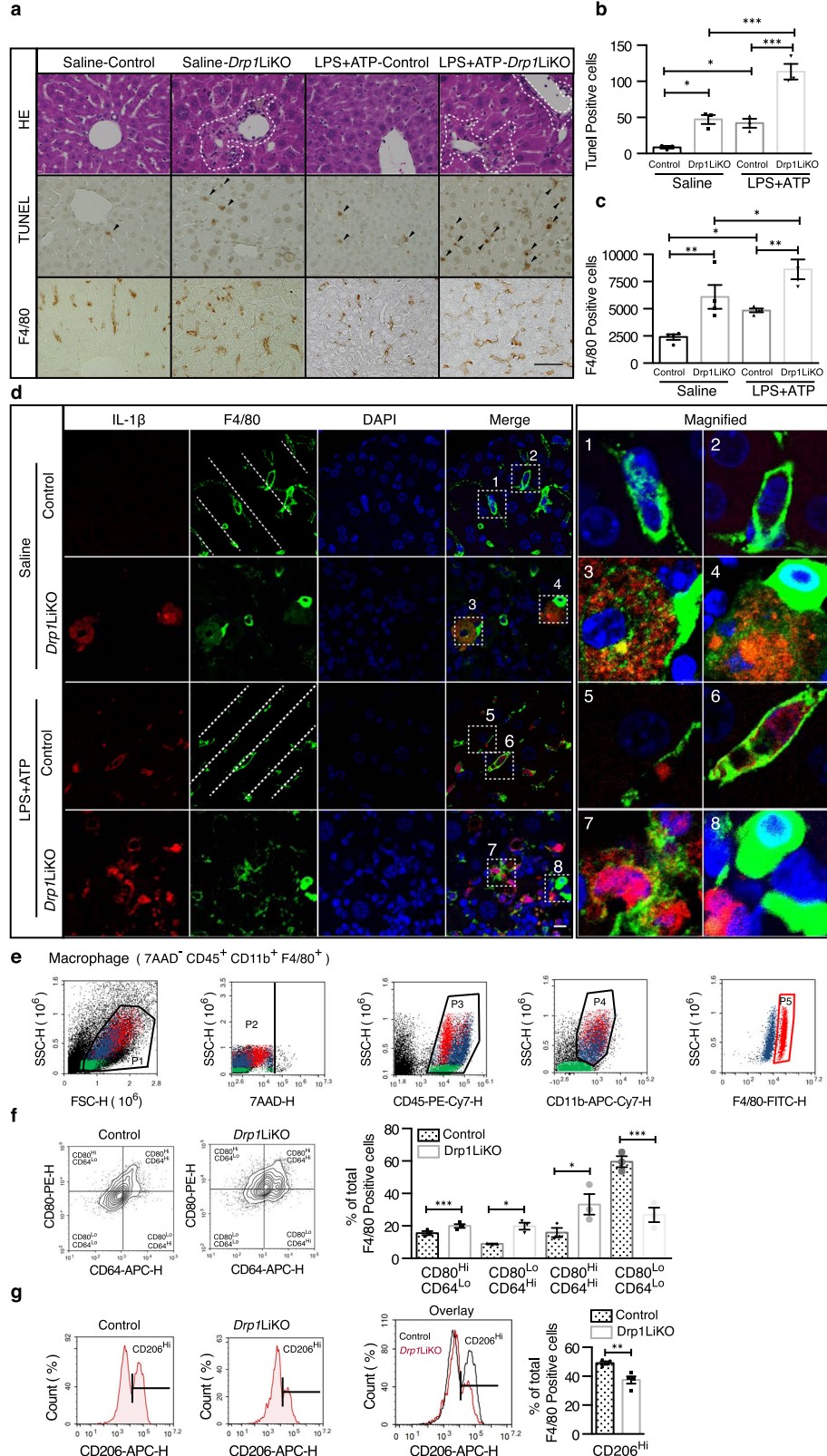

680 upregulated and 559 downregulated genes in *Drp1*LiKO cells. We used the DAVID tool for functional clustering of upregulated and downregulated genes via similarly annotated GO biological process terms. The top five enriched annotation clusters of upregulated genes are shown in Fig. 5b. All five clusters for upregulated genes were related to the immune system: the immune system process, defense response to virus, innate

immune response, response to virus, and immune response. Genes involved in the immune system process are shown in Supplementary Table 1. The most significant five clusters for downregulated genes were related to mitotic nuclear division, cell cycle, cell division, chromosome segregation, and mitotic chromosome condensation. Genes involved in mitotic nuclear division are shown in Supplementary Table 2. These data are

**Fig. 3 LPS/ATP-induced increased hepatocyte apoptosis and macrophage activation in *Drp1*LiKO mice. a–d** Control and *Drp1*LiKO mice were treated with LPS (10 mg/kg body weight, intraperitoneally) for 4 h, followed by an additional 50 μL intraperitoneally injection of 100 mM ATP 30 min before euthanization. H&E staining, TUNEL assay, and F4/80 immunostaining were performed on saline- or LPS/ATP-treated liver sections to examine histology (**a**). Areas with inflammatory cell infiltration are outlined by white dashed lines, and TUNEL-positive cells are indicated by arrowheads. Scale bar = 50 μm. **b** Quantification of TUNEL-positive cells in saline- or LPS/ATP-treated control and *Drp1*LiKO mice. Values are expressed as mean ± SEM. n = 3. *p < 0.05, ***p < 0.001 determined by Two-way ANOVA with Tukey's Post Hoc Test. **c** Quantitative analysis of the number of F4/80-positive cells performed by counting cells in 15 high-power fields (20×) per slide from 4 mice per group. Values are expressed as mean ± SEM. n = 4. *p < 0.05, **p < 0.01 determined by Two-way ANOVA with Tukey's Post Hoc Test. **d** Representative images of IL-1β and F4/80 staining of liver tissue from saline- or LPS/ATP-treated mice. IL-1β-positive cells were visualized using Alexa Fluor 594–conjugated chicken anti-goat IgG (red) and F4/80-positive cells using Alexa Fluor 488–conjugated rabbit anti-rat IgG (green). Nuclei were stained with DAPI (blue). Localization of Kupffer cells in the hepatic sinusoid is indicated (white dashed lines). Areas indicated with white dashed squares are enlarged and shown on the right side of the picture. Scale bar = 10 μm. **e–g** Drp1 ablation increases inflammatory macrophage infiltration in *Drp1*LiKO mice. **e** Gating strategy of macrophage (7AAD⁻CD45⁺CD11b⁺ F4/80⁺). Cell populations were gated sequentially from left to right. **f** Flow cytometry representative images and percentage analysis of pro-inflammatory macrophages isolated from the liver tissue of normal chow diet-fed mice using inflammatory macrophage markers CD64 and CD80. Values are expressed as mean ± SEM (n = 3). *p < 0.05 and ***p < 0.001 determined by an unpaired t test. **g** Flow cytometry representative images and percentage analysis of anti-inflammatory macrophages isolated from the liver tissue of normal chow diet-fed mice using anti-inflammatory macrophage marker CD206. Values are expressed as mean ± SEM. n = 4. **p < 0.01 determined by an unpaired t test. LPS lipopolysaccharide, ATP adenosine triphosphate, *Drp1*LiKO liver-specific *Drp1*-knockout, H&E hematoxylin and eosin, TUNEL terminal deoxynucleotidyl transferase dUTP nick end labeling, F4/80 adhesion G protein-coupled receptor E1, IL interleukin, DAPI 4′,6-diamidino-2-phenylindole, SEM standard error of the mean, IgG immunoglobulin G.

consistent with previous studies that reported that DRP1 loss leads to cell arrest, replication stress, centrosome overduplication, and chromosomal instability, subsequently increasing DNA damage and cell apoptosis[41]. Overall, these findings showed that *Drp1* deletion increases the expression of numerous genes involved in the immune response and induces DNA damage in primary hepatocytes.

We further compared mitochondrial morphology using Mito Tracker Red, a specific mitochondria fluorescence probe (Fig. 5c). In control mouse primary hepatocytes, 57.17% ± 4.00% of the mitochondria were fragmented, with only 14.80% ± 2.35% hepatocytes showing tubular morphology. *Drp1*LiKO mouse primary hepatocytes containing fragmented mitochondria decreased to 10.50% ± 2.66%, while those containing tubular mitochondria increased to 65.70% ± 3.77% (Fig. 5d). These results indicated less mitochondrial fission and more mitochondrial fusion in *Drp1*LiKO mouse primary hepatocytes compared to control mouse primary hepatocytes.

The mitochondrial respiratory chain generates a membrane potential across the mitochondrial inner membrane as protons are pumped across the inner membrane. $\Delta\Psi_m$ regulates matrix configuration and cytochrome *c* release during apoptosis[42]. Tetramethylrhodamine ethyl ester (TMRE) accumulates in the mitochondrial matrix based on $\Delta\Psi_m$ (Fig. 5e); the percentage of hepatocytes showing collapse of $\Delta\Psi_m$ were increased in the *Drp1*LiKO hepatocyte compared to the control mouse primary hepatocyte culture (Ctrl 17.68% ± 1.45% *vs*. KO 42.46% ± 3.88%), indicating that *Drp1* defect induced the loss of $\Delta\Psi_m$ (Fig. 5f).

**Decreased mitophagy formation and increased inflammatory response in LPS-treated *Drp1*LiKO mouse primary hepatocytes.** We further performed in vitro studies in which mouse primary hepatocytes were treated with LPS. Optic atrophy 1 (OPA1) is a mitochondrial fusion protein that resides in the inner mitochondrial membrane; co-localization of OPA1 and LC3 indicates the sequestration of mitochondria inside autophagosomes. Co-localization of Opa1 and LC3Lc3 was barely observed in *Drp1*LiKO mouse primary hepatocytes (Fig. 6a); thus, we measured the co-localization frequency of Opa1 and LC3Lc3, and found that this co-localization frequency decreased in *Drp1*LiKO mouse primary hepatocytes, which confirmed our finding that *Drp1* defects induce mitophagy defects (Fig. 6b).

We found that LPS induced a decrease in phosphorylation of S635 at 4 h and 8 h, whereas total Drp1 protein levels were

unchanged in control mice (Fig. 6c and d). Phosphorylation of S635 emerged before LPS treatment, which might have been due to a metabolic effect, as mitochondria were shown to be short and fragmented in control primary hepatocytes incubated in sustained high glucose conditions (Fig. 5c). We also detected significantly increased TNF, IL-6, and IL-10 levels in *Drp1*LiKO hepatocyte culture supernatants (Fig. 6e). Consistent with these observations, we found a significant increase in *Tnfa*, *Il6*, and *Il1b* mRNA expression in *Drp1*LiKO mouse primary hepatocytes compared to control mouse primary hepatocytes (Fig. 6f).

**ER stress induces an inflammatory response in *Drp1*LiKO mouse primary hepatocytes.** To validate our results in mice, we used mouse primary hepatocytes to investigate whether Drp1 deficiency induces inflammation through ER stress. We began by assessing the response to the well-known ER stress inducer thapsigargin. As thapsigargin concentration was increased, a dose-dependent increase in the expression of ER stress markers was observed; the expression of these markers was increased in thapsigargin-treated *Drp1*LiKO hepatocytes (Fig. 7a). Next, we compared the inflammatory responses to thapsigargin in control and *Drp1*LiKO hepatocytes. As expected, a significant increase in TNF and IL-6 protein levels was observed in the *Drp1*LiKO hepatocyte culture supernatant after thapsigargin treatment (Fig. 7b). Tauroursodeodeoxycholic acid (TUDCA) is an endogenous hydrophilic tertiary bile acid produces in humans at a low level. Mechanistic studies indicate that TUDCA prevents unfolded protein response dysfunction to attenuate ER stress[43]. Recent studies have revealed that TUDCA, as a classic ER stress inhibitor, has an ameliorating effect on several diseases, including neurodegenerative diseases, biliary cirrhosis, and cholestatic liver diseases[44,45]. We found that the mRNA expression levels of *Tnfa* and *Il6* were strongly inhibited by TUDCA treatment both in control and *Drp1*LiKO mouse primary hepatocytes (Fig. 7c). V-ZAD-FMK is a cell-permeable caspase inhibitor that has been shown to inhibit the induction of apoptosis[46,47]. In the present study, treatment with V-ZAD-FMK also reduced the mRNA expression levels of *Tnfa* and *Il6* (Fig. 7c).

**Discussion**

Many studies have reported the abnormal function of mitochondrial dynamics in liver disease. Although mitochondrial fusion defects in the liver cause a NASH-like phenotype and liver

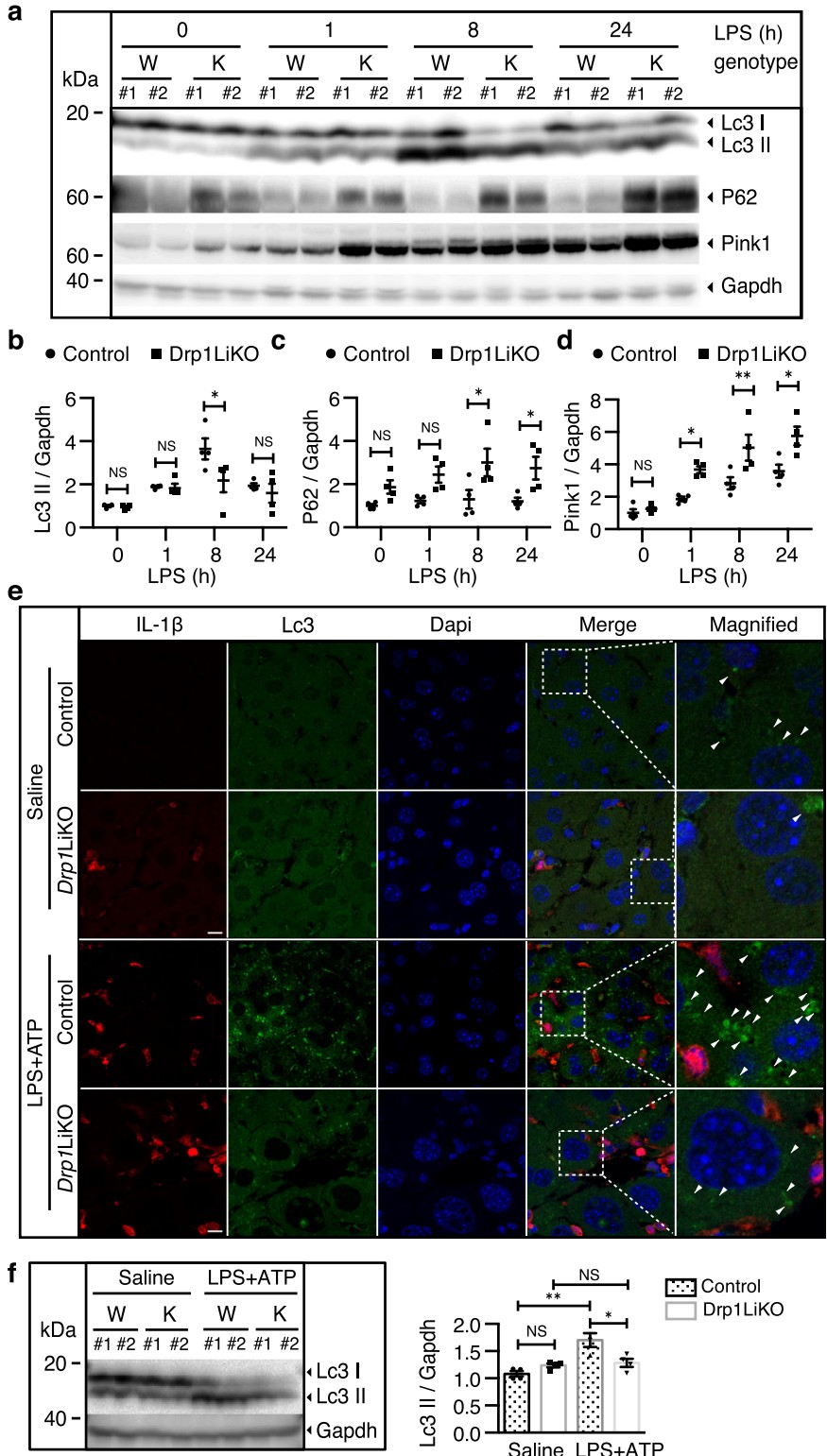

cancer, whether mitochondrial fission defects directly impair liver function and stimulate liver disease progression is unclear. In the present study, Drp1 defects led to elongated mitochondria with $\Delta\Psi_m$ collapse, mitophagy defect, DNA damage, and high expression of the ER stress-related genes *P8*, *Chop*, and *Trib3*. In turn, these changes increased hepatocyte death in *Drp1*LiKO mice and the dead hepatocytes released danger signals that led to inflammatory macrophage infiltration (Fig. 7d). Furthermore,

Drp1 loss in the liver led to an accelerated inflammatory response induced by LPS. Compared with control mice, *Drp1*LiKO mice showed increased expression of pro-inflammatory cytokines in the liver and serum, increased infiltration of inflammatory macrophages, enhanced inflammasome activation in the liver, increased hepatocyte apoptosis, and decreased mitophagy formation in hepatocytes. ER stress pathway protein-folding and quality control functions maintain cell homeostasis and are

**Fig. 4 Drp1 defect decreased autophagy in *Drp1*LiKO mice. a–d** Control and *Drp1*LiKO mice were treated with saline alone (referred to as LPS 0H) or LPS (5 mg/kg, intraperitoneally). At indicated time points after LPS treatment, liver tissue was collected. Liver Lc3, P62, and Pink1 levels were determined by western blot analysis. **b–d** Densitometric quantification of Lc3, P62, and Pink1 expression in saline- or LPS-treated liver lysates. Gapdh served as an internal control. The biological duplicate samples in each condition numbered as #1 and #2. Values are expressed as mean ± SEM. *n* = 4. *p < 0.05, **p < 0.01 determined by Two-way ANOVA with Bonferroni's Post Hoc Test. **e** Representative images of IL-1β and Lc3 staining of liver tissue from LPS/ATPSaline- or LPS/ATP-treated-treated mice. IL-1β-positive cells were visualized using Alexa Fluor 594-conjugated chicken anti-goat IgG (red) and Lc3 positive cells using Alexa Fluor 488-conjugated donkey anti-rabbit IgG (green). Nuclei were stained with Dapi (blue). Areas indicated with white dashed squares are enlarged and shown on the right side of the picture. Arrowheads indicate Lc3-positive dots in hepatocytes. Scale bar = 10 μm. **f** Western blot analysis of Lc3 and Gapdh expression in saline- or LPS/ATP-treated liver lysates. The bar graph shows the densitometric quantification of Lc3-II. Gapdh served as an internal control. The biological duplicate samples in each condition numbered as #1 and #2. Values are expressed as mean SEM. *n* = 4. *p < 0.05, **p < 0.01 determined by Two-way ANOVA with Tukey's Post Hoc Test. *Drp1*LiKO liver-specific *Drp1*-knockout, LPS lipopolysaccharide, Sqstm1 sequestosome-1, Pink1 phosphatase and tensin homolog-induced putative kinase 1, Gapdh glyceraldehyde 3-phosphate dehydrogenase, IL interleukin, Lc3 light chain 3, ATP adenosine triphosphate, IgG immunoglobulin G, NS no significant difference, SEM standard error of the mean, Dapi 4′,6-diamidino-2-phenylindole.

closely linked to mechanisms underlying immunity and inflammation[25,48]. First, we found that exaggerated ER stress occurred in the LPS-treated liver of *Drp1*LiKO mice and in thapsigargin-treated *Drp1*LiKO mouse primary hepatocytes. Second, we found that the expression levels of inflammatory cytokines were increased as a direct consequence of this elevated ER stress response. Third, TUDCA treatment dramatically reduced the expression levels of inflammatory cytokines, exhibiting anti-inflammatory properties. Thus, the ER stress pathway is evoked in *Drp1*LiKO mice to promote hepatocyte apoptosis and liver inflammation.

Overall, these findings show that, upon an LPS challenge, exaggerated ER stress response in hepatocytes and NLRP3 inflammasome overactivation in inflammatory macrophages jointly contribute to the elevated inflammatory response observed in *Drp1*LiKO mice. Our study reveals that hepatocytes lacking DRP1 have an increased inflammatory response, whereas several recent studies have shown that a lack of DRP1 leads to lower inflammatory responses in macrophages and microglia[17,18]. Hepatocytes and macrophages might be physiologically different in terms of DRP1-dependent apoptosis. Therefore, it will be worth studying and clarifying the function of DRP1 in apoptosis and inflammation according to cell type. In conclusion, using an acute liver injury experimental mouse model produced via LPS-induced endotoxin shock, we have presented approaches to studying DRP1 defect-induced liver inflammation progression that will provide insights into the role played by DRP1 in liver function and acute liver injury.

Drp1 deletion leads to the loss of $\Delta\Psi_m$ in *Drp1*LiKO mouse primary hepatocytes; since loss of $\Delta\Psi_m$ is closely linked to cell apoptosis by various insults, we believe that the increased population with $\Delta\Psi_m$ collapse in *Drp1*LiKO mouse primary hepatocytes might be an insult and/or a consequence induced by increased cell apoptosis. We have previously reported that, despite the observed morphological changes in mitochondria, the expression levels of the mitochondrial respiratory chain complex and mitochondrial respiratory activity were not changed by loss of DRP1[14]. Opening of the mitochondrial permeability transition pore (mPTP) causes inner membrane potential collapse and eventually leads to mitochondrial swelling, rupture, and cell death. In DRP1-deficient hearts and MEFs, inhibition of DRP1-mediated fission produces elongated mitochondria with increased mPTP opening[49]. Furthermore, accelerated mPTP opening in DRP1 null mitochondria has been associated with mitophagy in MEFs and contributes to cardiomyocyte necrosis and dilated cardiomyopathy in mice[50]. Mitochondrial outer membrane permeabilization (MOMP) also induces the loss of $\Delta\Psi_m$ and results in the release of apoptotic proteins that activate the downstream pathway of apoptosis. Although the mechanism by which DRP1 participates in MOMP remains to be elucidated, previous studies

indicate that DRP1 inhibition has direct effects on MOMP[51,52] and that the role of DRP1 in MOMP can be distinguished from mitochondrial fission[53]. Several studies have shown that $\Delta\Psi_m$ increases when mitophagy or autophagy are inhibited[54–56]; nevertheless, it is possible to conclude that the increased population observed with $\Delta\Psi_m$ collapse in *Drp1*LiKO mouse primary hepatocytes may have been caused by mPTP opening and/or MOMP, which are independent of mitochondrial fission or mitophagy formation.

Mitochondrial dynamics and mitophagy have gained significant interest because these events modulate mitochondrial function during many physiological and pathological conditions. In a mouse embryonic fibroblast model, *Drp1* mutant cells showed abnormal mitochondrial morphology and defective mitophagy, leading to the activation of sterile myocardial inflammation, resulting in heart failure[57]. In INS1 cells, inhibition of the fission machinery through a dominant-negative form of DRP1 (DRP1[K38A]) decreases mitochondrial autophagy and results in the accumulation of damaged mitochondrial material[58]. In a transverse aortic constriction mouse model, DRP1-dependent mitochondrial autophagy plays a protective role against pressure overload–induced mitochondrial dysfunction and heart failure[59]. In addition, Parkin-independent mitophagy requires DRP1 and maintains the integrity of the mammalian heart and brain[60]. Insufficient mitophagy formation induced by DRP1 defects might increase the inflammatory response and enhance the severity of liver injury in response to multiple pathogens.

Hepatitis, an inflammatory condition of the liver, is a serious global public health problem. Although it is usually caused by a viral infection, several other possible risk factors exist, including infections, alcohol, toxins, drugs, and autoimmune diseases. In this study, we demonstrated that mitochondrial fission defects caused by the lack of DRP1 lead to tissue damage associated with hepatitis, which suggests that DRP1 could be a therapeutic target for inflammatory liver disease. Under acute inflammatory liver disease, DRP1 could be potentially activated to restore a protective fission in hepatocytes. In recent years, lipid nanoparticles have been developed to passively and actively target drugs to the liver. Liver-targeted lipid nanoparticles with the expression of DRP1 or ablation of MFN2 by siRNA treatment (thereby inducing fission) could be used to produce anti-inflammatory activities. Overall, our study reveals that mitochondrial fission defects directly impair liver function and stimulate liver disease progression. Our examination of *Drp1*LiKO mice uncovered the essential role of mitochondrial fission in mitochondrial quality control for the prevention of hepatic inflammation. This exciting discovery offers a promising alternative approach to develop therapeutic strategies for the treatment of inflammatory liver disease.

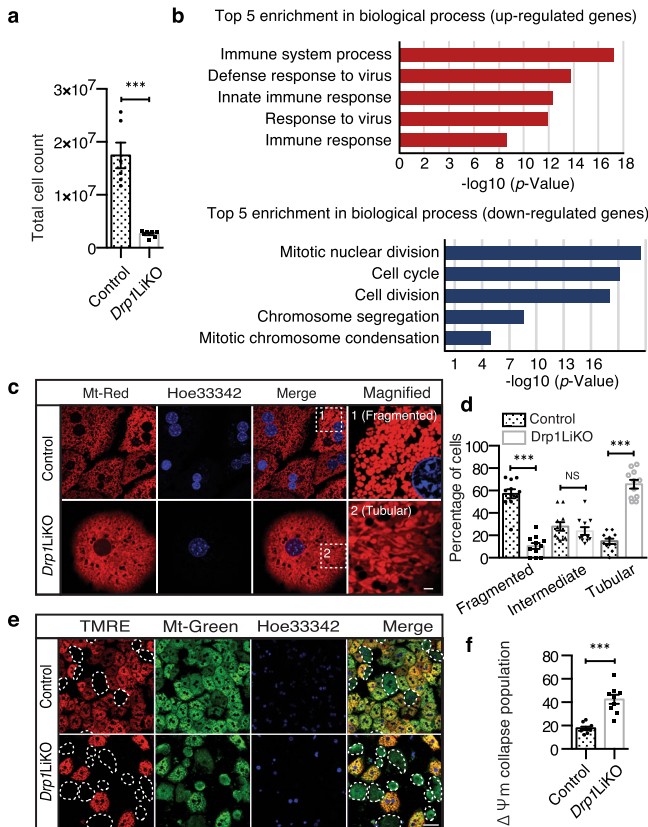

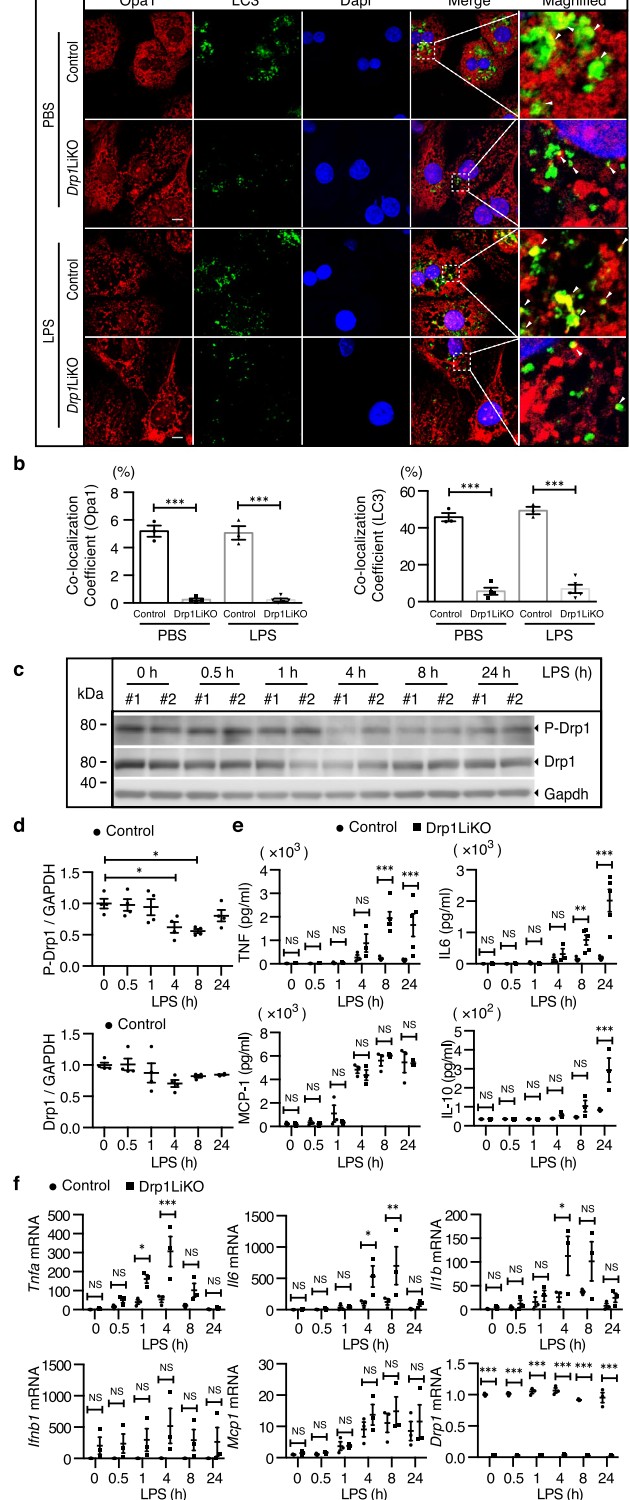

**Fig. 5 Microarray functional annotation analysis, mitochondrial morphology, and mitochondrial membrane potential ($\Delta\Psi_m$) in control and *Drp1*LiKO mouse primary hepatocytes. a** Total mouse primary hepatocyte numbers from control and *Drp1*LiKO mice were measured by trypan blue staining to measure viable cells. $n = 6\text{-}7$. ***$p < 0.001$ determined by an unpaired $t$ test. **b** Top five biological process GO terms of up- and down- regulated genes ranked by $p$ value. **c, d** To visualize mitochondria, control and *Drp1*LiKO mouse primary hepatocytes were stained with MitoTracker® Red. Nuclei were stained with Hoechst33342, and images were captured with a confocal microscope. Areas indicated with white dashed squares are enlarged and shown on the right. Mitochondrial morphology was graded as fragmented, intermediate, or tubular for control and *Drp1*LiKO mouse primary hepatocytes. Quantification was performed by counting cells in 3–5 high-power fields (60×) per slide from three independent experiments. Scale bar = 10 μm. ***$p < 0.001$ determined by an unpaired $t$ test. **e, f** Mitochondrial membrane potential ($\Delta\Psi_m$) measured in control and *Drp1*LiKO mouse primary hepatocytes using the fluorescence probe TMRE. Mitochondria and nuclei were stained by MitoTracker® Green and Hoechst33342, respectively. White dashed lines track cell boundaries with dissipated $\Delta\Psi_m$ (collapsed $\Delta\Psi_m$). Quantification of $\Delta\Psi_m$ collapse population was performed by counting cells in three high-power fields (20×) per slide from three independent experiments. Scale bar = 50 μm. ***$p < 0.001$ determined by an unpaired $t$ test. *Drp1*LiKO liver-specific *Drp1*-knockout, SEM standard error of the mean, GO Gene Ontology, TMRE tetramethylrhodamine ethyl ester perchlorate.

## Methods

**Animals and reagents**. We purchased 8–12-week-old male C57BL/6 J mice from KBT Oriental Co., Ltd. (Saga, Japan). *Drp1*LiKO (*Alb-Cre^{Tg/+}^ Drp1^{flox/flox}^*) and control (*Drp1^{flox/flox}^*, sibling littermates) mice were generated by crossing *Drp1^{flox/+}^* mice[14] and *Alb-Cre* mice. The mice were maintained in a standard specific-pathogen free room at room temperature (22 °C–24 °C) and 50–60% relative humidity under a 12 h/12 h light/dark cycle (lights off at 8:00 p.m.). The mice were fed a normal chow diet (NCD; 5.4% fat, CRF-1; Orient Yeast, Tokyo, Japan) *ad libitum*. Experiments were performed between 10:00 and 11:00 a.m. after overnight fasting, except the LPS challenge and primary hepatocyte isolation (ad libitum). All

mouse procedures and protocols were approved by the Ethics Committees on Animal Experimentation (Kyushu University, Graduate School of Medicine, Japan) and performed in accordance with the Guide for the Care and Use of Laboratory Animals.

Reagents used in this study were listed in Supplementary Table 3.

**Mouse primary hepatocyte isolation and culture**. We performed a rapid two-step method[61] to isolate mouse primary hepatocytes. We anesthetized the mice with pentobarbital sodium (50 mg/kg body weight) and perfused their liver tissue with prewarmed Hank's balanced salt solution supplemented with 0.5 mM ethylene glycol tetraaceticacid (EGTA) and 10 mM 4-(2-hydroxyethyl)-1-piperazineethanesulfonic

**Fig. 6 Decreased mitophagy formation and increased inflammatory response induced by LPS treatment in *Drp1*LiKO mouse primary hepatocytes.** Control and *Drp1*LiKO mouse primary hepatocytes were treated with PBS (referred to as LPS 0 h) or LPS (100 ng/ml). At indicated time points after LPS treatment, cells and culture supernatants were collected. **a** Representative images of Opa1 and Lc3 staining of PBS or LPS-treated (24 h) control and *Drp1*LiKO mouse primary hepatocytes. Opa1-positive cells were visualized using Alexa Fluor 594-conjugated goat anti-mouse IgG (red) and Lc3-positive cells using Alexa Fluor 488-conjugated donkey anti-rabbit IgG (green). Nuclei were stained with Dapi (blue). Areas indicated with white dashed squares are enlarged and shown on the right side of the picture. Scale bar = 10 μm. **b** Statistical analysis of the co-localization efficiency of Opa1 and Lc3 by using ZEN software. $n = 3$. ***$p <$ 0.001 determined by Two-way ANOVA with Tukey's Post Hoc Test. **c, d** Western blot analysis and densitometric quantification of p-Drp1(S635) and Drp1 in PBS or LPS-treated cells. Gapdh served as an internal control. The biological duplicate samples in each condition numbered as #1 and #2. Values are expressed as mean ± SEM ($n = 4$). *$p < 0.05$ determined by one-way ANOVA with Dunnett's multiple comparisons test. **e** TNF, IL-6, MCP1, and IL-10 levels were determined using a BD cytometric bead array ($n =$ 3–5). **$p < 0.01$ and ***$p < 0.001$ determined by two-way ANOVA with Bonferroni's post hoc test. **f** Total RNA was isolated at indicated time points after LPS treatment, and *Tnfa*, *Il6*, *Il1b*, *Ifnb1*, *Mcp1*, and *Drp1* expression was determined by quantitative real-time PCR. Results are normalized to *Gapdh* expression and are shown as fold-changes relative to gene expression in untreated control cells. Values are expressed as mean ± SEM ($n = 3$). *$p <$ 0.05, **$p < 0.01$, and ***$p < 0.001$ determined by two-way ANOVA with Bonferroni's post hoc test. *Drp1*LiKO liver-specific *Drp1*-knockout, LPS lipopolysaccharide, Opa1 optic atrophy 1, Lc3 light chain 3, IgG immunoglobulin G, Dapi 4′,6-diamidino-2-phenylindole, IL interleukin, TNF tumor necrosis factor, MCP1 monocyte chemoattractant protein-1, IFN-γ interferon gamma, *Gapdh* glyceraldehyde 3-phosphate dehydrogenase, SEM standard error of the mean.

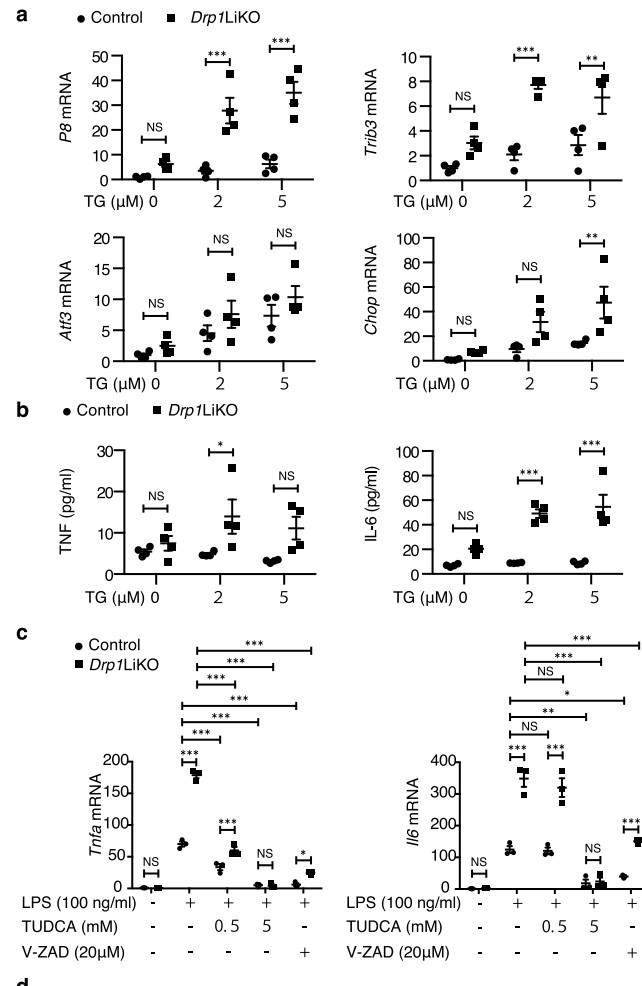

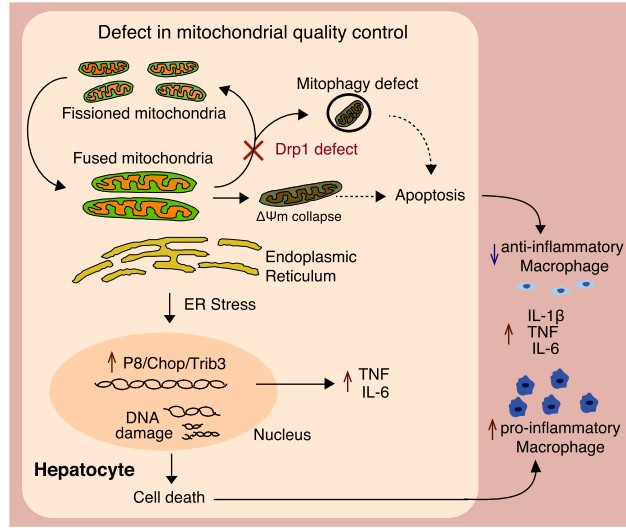

acid (HEPES) for 5 min, followed by digestion buffer (1000 mg/L of low-glucose Dulbecco's modified Eagle's medium [DMEM] supplemented with 0.8 mg/mL of collagenase type 1; Worthington Biochemical Corporation, Lakewood, NJ, USA). Next, we filtered single-cell suspensions through a 70-μm cell strainer (BD Falcon, Bedford, MA, USA) and centrifuged them at $50 × g$ for 1 min and collected the cell supernatants separately for the preparation of nonparenchymal cells, as described later. We washed the hepatocytes in the pellet twice, suspended them in 4500 mg/L of high-glucose DMEM supplemented with 1 μM insulin, 2 mM L-glutamine, 10 IU/mL of penicillin, 10 IU/mL of streptomycin, and 10% fetal bovine serum, and seeded them on 6-well plastic plates coated with collagen type I (Cellmatrix type I-P, Nitta Gelatin, Osaka, Japan) at a density of 300,000 cells/well. After 4 h incubation in a 5% $CO_2$ incubator at 37 °C, we changed the culture medium to discard floating unattached cells, and the growth medium was changed daily.

**Mouse hepatic nonparenchymal cell isolation and flow cytometry**. To further explore the polarization signature of macrophages in the liver of control and *Drp1*LiKO mice, we performed flow cytometry assay and sequential gating analysis using isolated nonparenchymal hepatocytes without LPS injection. We prepared hepatic nonparenchymal cell containing suspensions from the liver, as described before. The suspensions were centrifuged at $50 × g$ for 5 min to remove the remaining hepatocytes, and the resulting cell suspensions were pelleted by centrifugation at $800 × g$ for 5 min and resuspended in phosphate-buffered saline (PBS) supplemented with 5% bovine serum albumin. Next, we stained cells with appropriate antibodies (Supplementary Table 4) for 30 min on ice, washed them, and analyzed them using a NovoCyte flow cytometer (ACEA Biosciences, San Diego, CA, USA) and NovoExpress software.

**Total RNA isolation, microarray procedures, and real-time-PCR**. We isolated total RNA from the mouse liver and primary hepatocytes using TRIzol Reagent (Invitrogen Corporation, Carlsbad, CA, USA) and purified it using the SV Total RNA Isolation System (Promega Corporation, Madison, WI, USA) according to the manufacturer's instructions. For microarray analysis, we amplified complementary RNA (cRNA) and labeled it using the Low input Quick Amp Labeling Kit (Agilent Technologies, Santa Clara, CA, USA). Next, we hybridized the cRNA to a 60 K 60-mer oligomicroarray (SurePrint G3 Mouse Gene Expression

Microarray 8x60K v2; Agilent Technologies) according to the manufacturer's instructions.

For real-time-PCR assays, we converted 500 ng of total RNA into first-strand complementary DNA (cDNA) using the QuantiTect Reverse Transcription Kit (QIAGEN, Hilden, Germany) according to the manufacturer's instructions. Next, we used the cDNA for quantitative real-time PCR using the power 2 × SYBR Green PCR Master Mix and monitored the process using an ABI Prism 7500 sequence detection system (Thermo Fisher Scientific, Rockford, IL, USA). Supplementary Table 5 lists the primer sequences of the selected genes. We normalized relative gene expression versus control to *Gapdh* expression.

**Fig. 7 Increased inflammatory response was inhibited by TUDCA pre-treatment in *Drp1*LiKO mouse primary hepatocytes. a, b** Thapsigargin induced an elevated ER stress response and an increased inflammatory response in *Drp1*LiKO mouse primary hepatocytes. Control and *Drp1*LiKO mouse primary hepatocytes were treated with PBS (referred to as TG 0 µM) or Thapsigargin (2 or 5 µM) for 24 h. **a** *Atf3*, *Chop*, *P8*, and *Trib3* mRNA expression were determined by quantitative real-time PCR. Results are normalized to *Gapdh* expression and are shown as fold-changes relative to gene expression in PBS-treated control hepatocytes. Values are expressed as mean ± SEM ($n = 3$-4). $**p < 0.01$ and $***p < 0.001$ determined by two-way ANOVA with Sidak's multiple comparisons test. **b** Culture supernatants were collected and TNF and IL-6 were determined by ELISA. Values are expressed as mean ± SEM ($n = 4$). $*p < 0.05$ and $***p < 0.001$ determined by two-way ANOVA with Sidak's multiple comparisons test. **c** Control and *Drp1*LiKO mouse primary hepatocytes were pre-treated with PBS (referred to as TUDCA 0 mM), TUDCA (0.5 or 5.0 mM) for 24 h, or V-ZAD-FMK (20 µM) for 1 h, and then cells were treated with PBS (referred to as LPS -) or LPS (100 ng/ml) for 4 h. Total RNA was isolated and *Tnfa* and *Il6* expression was determined by quantitative real-time PCR. Results are normalized to *Gapdh* expression and are shown as fold-changes relative to gene expression in untreated control cells. Values are expressed as mean ± SEM ($n = 3$). $**p < 0.01$ and $***p < 0.001$ determined by two-way ANOVA with Tukey's post hoc test. **d** Schematic model of liver inflammation in *Drp1*LiKO mice. *Atf3* activating transcription factor 3, *Chop* DNA damage–inducible transcript 3 (*Ddit3*), *P8* nuclear protein 1 (*Nupr1*), *Trib3* Tribbles homolog 3, TNF tumor necrosis factor, IL interleukin, PCR polymerase chain reaction, *Gapdh* glyceraldehyde 3-phosphate dehydrogenase, ER endoplasmic reticulum, *Drp1*LiKO liver-specific *Drp1*-knockout, LPS lipopolysaccharide.

**Western blot analysis**. We homogenized fresh mouse liver tissue in western lysis buffer (20 mM Tris-HCl pH 7.6, 150 mM NaCl, 2 mM ethylenediaminetetraacetic acid [EDTA], 0.5% NP40) containing protease and phosphatase inhibitor tablet (Roche Applied Science, Mannheim, Germany) and centrifuged it at $10,000 \times g$ for 5 min at 4 °C. We discarded the pellet and measured the protein content using a bicinchoninic acid (BCA) Protein Assay kit (Thermo Fisher Scientific). Next, the samples were mixed with Laemmli sample buffer (1:1; Bio-Rad, Hercules, CA, USA) and heated for 5 min at 95 °C. We performed sodium dodecyl sulfate polyacrylamide gel electrophoresis (SDS-PAGE) on an equal amount of protein from each sample, trans-blotted it on a polyvinylidene difluoride (PVDF) membrane, and subjected it to immunoblot assay with primary antibodies, followed by horseradish peroxidase (HRP)-linked secondary antibodies. Supplementary Table 4 lists the primary and secondary antibodies used. Finally, we visualized protein bands using the electrochemiluminescence (ECL) Western Blotting Detection System (GE Healthcare, Buckinghamshire, UK).

**Histology, immunohistochemistry, and live-cell imaging**. We fixed liver tissue and primary hepatocytes by immersion in 4% (w/v) paraformaldehyde (PFA) overnight at 4 °C, embedded the liver tissue in paraffin according to standard methods, and cut it into 5-µm-thick sections. For H&E staining, we stained the paraffin sections with Mayer's hematoxylin solution for 5 min, followed by counterstaining with 0.5% (v/v) eosin alcohol solution for 2 min. For immunostaining, we diluted primary and secondary antibodies in Can Get Signal™ Immunoreaction Enhancer Solutions A and B, respectively (Toyobo, Osaka, Japan). Supplementary Table 4 lists the primary and secondary antibodies used. Next, we analyzed hepatocyte apoptosis using the TUNEL assay kit (Burlington, Boston, MA, USA) according to the standard protocol. The tissue sections were analyzed under a BZ-8000 microscope (Keyence, Osaka, Japan) or a confocal microscope LSM700 (Zeiss, Oberkochen, Germany). The OPA1 and LC3 co-localization measurements were performed using ZEN software (Zeiss, Oberkochen, Germany) according to the manufacturer's instructions.

For live-cell imaging, we seeded hepatocytes on a type I-coated 35 mm glass-based dish (Iwaki, Tokyo, Japan) preloaded with MitoTracker® Red, a specific mitochondria fluorescence probe. We further measured $\Delta\Psi_m$ using a fluorescence probe tetramethylrhodamine ethyl ester perchlorate (TMRE). Next, we examined the hepatocytes under the confocal microscope LSM700 and analyzed image data using ZEN software (Zeiss, Oberkochen, Germany). A hepatocyte was judged to have fragmented mitochondria if <25% of the mitochondria visible had a length five times their width and tubular if >75% of the mitochondria had a length five times their width.

**Measurement of cytokine levels in the liver, serum, and cell culture supernatants**. LPS was injected intraperitoneally into mice at 5 mg/kg body weight, and liver and serum samples were collected from the inferior vena cava at different time points (0, 1, 4, 8, 24, and 48 h). The liver tissue was perfused with cold PBS through the portal vein and then chopped into 1-2 mm pieces. Next, we added 100 mg of liver tissue to 1 mL of cell lysis buffer (R&D System, Minneapolis, MN, USA) and homogenized it using a tissue homogenizer. Finally, we diluted liver, serum, and cell culture supernatant samples with known high protein concentrations at 1:50 using an assay diluent and quantified the levels of various types of cytokines using the BD cytometric bead array (CBA) Cytokine kit (Becton-Dickinson, Franklin Lakes, NJ, USA) and a NovoCyte flow cytometer (ACEA Biosciences, San Diego, CA, USA) and NovoExpress software.

**Biochemical assays**. To evaluate the degree of functional damage in the liver, we measured serum ALT and AST levels (two markers of hepatocellular injure or necrosis) using the DRI-CHEM3500 Chemistry Analyzer (Fujifilm, Tokyo, Japan).

**Enzyme-linked immunosorbent assay**. Primary mouse hepatocytes were treated with thapsigargin (final concentration: 2 or 5 µM) and the culture supernatants were collected and centrifuged to remove nonadherent cells. The concentrations of mouse TNF (Ab208348; Abcam, Cambridge, MA) and mouse IL-6 (M6000B; R&D Systems, Inc., Minneapolis, MN) were determined according to manufacturer's instructions. The serum and liver gasdermin D levels were determined using Gasdermin D (mouse) ELISA Kit (AG-45B-0011-K101; AdipoGen Life Science, Switzerland) according to manufacturer's instructions.

**Statistics and reproducibility**. All data were expressed as the mean ± standard error of the mean (SEM). Two-tailed Student's $t$ test was performed to compare two groups using Microsoft Excel (Mac 201012 version 16.16.27; Microsoft Japan, Tokyo, Japan). Two-way analysis of variance (ANOVA) with Bonferroni's Post Hoc Test or Tukey's post hoc test or ordinary one-way ANOVA was performed to compare multiple groups using GraphPad Prism 6.0 software (GraphPad, San Diego, CA, USA). To detect an outlier, we performed Grubbs' test using a free statistical calculator (QuickCals, http://www.graphpad.com/quickcalcs/). Significance levels were set at $p < 0.05$, $p < 0.01$, and $p < 0.001$.

**Reporting summary**. Further information on research design is available in the Nature Research Reporting Summary linked to this article.

## Data availability
The microarray data from this publication have been submitted to the Gene Expression Omnibus database [http://www.ncbi.nlm.nih.gov/geo/] with an assigned identifier [accession: GSE156982]. Source data generated or analyzed during this study are provided in Supplementary Data 1 (source data for graphs) and a Supplementary information file (Supplementary Figs. 1–4, Supplementary Tables 1–7 and uncropped western blot images). The data that support the findings of this study are available from the corresponding authors upon reasonable request. A reporting summary for this article is available as a supplementary information file.

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

## Acknowledgements

We appreciate the technical assistance from The Research Support Center, Research Center for Human Disease modeling, Kyushu University Graduate School of Medical Sciences. This work was supported in part by the Japanese Society for the Promotion of Science (JSPS) KAKENHI (M. Nomura, Grant Number 23591356; L. Wang, Grant-in-Aid for Young Scientists B, Grant Number 16K19557).

## Author contributions

L.W. and M.N. designed the experiments. L.W. performed the experiments. Y.H. and N.H. supported experiments. L.W. analyzed the data and wrote the manuscript. M.N., X.L., K.Y. and Y.M. supervised the analysis and edited the manuscript. M.N. and K.Y. are the guarantors of the article.

## Competing interests

The authors declare no competing interests.
