## [Peer Review File · Communications Biology]

Reviewers' Comments:

Reviewer #1:

Remarks to the Author:

In their study entitled "Dynamin-related 1 protein 1 deficiency impairs mitophagy and accelerates lipopolysaccharide-induced inflammation in mice, Wang et al. investigate the role of DRP1, a key protein in the control of mitochondrial dynamics, in liver inflammation.

This is a very interesting study. Several points need to be improved and/or clarified.

Major points:

The authors performed LPS-injection to generate liver dysfunction. It would be interesting to assess the impact of liver DRP1 deficiency in an additional model such as acetaminophen-induced liver dysfunction or in the context of bacterial infection (e.g IP injection of E. Coli or CLP-induced bacterial infection). Indeed, the authors showed an increase in M1 macrophages in the liver. Would it translate into a reduced bacterial load in the live as the inhibition of mitophagy in macrophages was recently shown to induce macrophage activation and to promote anti-bacterial defense (Patoli et al., JCI, Oct 2020) ?

Experimental conditions should be more clearly indicated on the figures. Indeed, it is not always clear whether mice or cells have been treated with LPS. In addition, the authors should systematically assess the same condition i.e WT, WT+LPS, liver KO, Liver KO + LPS. Conditions without LPS exposure are not always depicted.

In addition to DRP1 mRNA levels in the liver of LPS-treated mice and isolated hepatocytes the authors should also assess DRP1 protein levels (total and phosphorylated forms).

What is the impact of liver DRP1 deficiency upon LPS treatment in term of mortality?

Infiltration of macrophages by immune cells should be assessed by flow cytometry and not only by microscopy approaches which are not sufficiently accurate for such a purpose. Why infiltration of the liver by immune cells was not assessed in LPS treated mice ? LPS + ATP is a very different model and NLRP3 is also activated upon LPS treatment alone.

The proportion of M1 macrophages should be assessed with classical markers: e.g CD80, CD86, CD64. Ly6C is a marker for monocytes. F4/80⁺-CD11b⁺-Ly6C⁺ cells do not allow to discriminate between monocytes and macrophages. It is surprising that all CD11b-F4/80 positive cells are expressing Ly6C at a similar level. Usually, several populations: Ly6 Low, Ly6C med and Ly6 High can be observed.

How the authors explain the decrease in autophagy in LiDRP1 KO mice treated with LPS ?

The authors propose that mitophagy is decreased in the hepatocytes of LPS-treated animals. This is consistent with previous studies. Nevertheless, how the authors explain the drop in the mitochondrial membrane potential ? Several studies showed that mitochondrial membrane potential is increased when mitophagy or autophagy are inhibited (Zhou R, et al Nature, 2011, 469(7329):221–225; Gomes LC, et al, Nat Cell Biol. 2011;13(5):589–598. Yao Z, et al J Cell Sci. 2011;124(Pt 24):4194–4202, Patoli et al., JCI, Oct 2020).

The authors claim that DRP1 deficiency promotes inflammation through ER stress (Fig 7). This is not demonstrated in the paper. Such a claim has to be supported by experimental data using KO mice or pharmacological modulation of ER stress in WT and LiDRP1 KO mice or hepatocytes exposed or not to LPS. Is LiDRP1 KO-mediated liver inflammation blunted when P8/Chop/trib3 is inhibited or invalidated? This point is critical as the data do not fully support the model proposed in the discussion.

Minor:

Statistical analysis should be depicted in the figure legends.

Fig 1a: Is there a difference in the basal levels of cytokines in WT and liver DRP1 KO mice ?

Fig 2b: The authors should normalize all mRNA levels to WT mice at t0 and not to the t0 for each genotype.

Why 7AAD is used for the characterization of M2 macrophages but not for M1 macrophages (since the goal is to exclude dead cells)?

Page 19 line 272 to 278. Explanation regarding mitophagy levels are not clear. This paragraph needs to be clarified as well as the corresponding figure which does not clearly display the experimental conditions.

Reviewer #2:

Remarks to the Author:

The manuscript by Wang et al. is investigating how the lack of Drp-1 in hepatocytes leads to an increased inflammation. These findings are timely as the role of mitochondrial dynamics in inflammatory diseases is under investigation by numerous research teams. This work uses multiples approaches to confirm their findings and is generally convincing. This is a interesting manuscript and the aims of this paper are clear. There are although some points that would require some clarification:

1- Apoptosis was measured by TUNEL but this type of cell death could be confirmed by other means (caspase 3 activation, cytochrome C activation)? In addition, can the author see M1 macrophages in close proximity to apoptotic hepatocyte in their microscopy?

2- Furthermore, the mechanism remains unclear on how the lack of Drp1 triggers apoptosis. The authors highlight that Trib3 is more expressed in the Drp1-LiKO cells, which could lead to apoptosis. Would inhibition of Trib3 (i.e siRNA) improve hepatocyte survival and decrease apoptosis level (caspase 3)? In the discussion, the authors suggest that ER-stress would increase ROS production, is that the case for those hepatocytes ? If so, does reducing ROS increases hepatocytes survival? Ultimately, does treatment with caspase inhibitor (such as z-VAD) could reduce the inflammation level in Drp1-LiKO mice ?

3- In Figure 5, the number of hepatocytes recovered is dramatically different between the two genotypes. Maybe as a result, it seems that the confluence of the cells is quite different also (by microscopy). How consistent was the cell confluence? Would a difference in the cell confluence have an impact on cellular stress (i.e TMRM)?

4- It might be worth discussing the unclear function of Drp1 in inflammation upon the cell type. The authors show that hepatocytes lacking Drp1 have an increased inflammatory response but several recent publications (PMID: 32686869, PMID: 30637805, PMID: 23815397) have also shown that macrophages lacking drp-1 have a lower inflammatory response. Are macrophages and hepatocyte physiology different in regard to Drp1-dependent apoptosis?

Minors comments:

1- Mitofusin and not Mitofusion (Line 43)

2- As there is an important difference in apoptosis (re the number of primary hepatocytes harvested), one can wonder if the liver of the Drp1-LiKO are physiologically perturbed. Indeed, in Fig 1c, one can see a small but constantly increases AST and ALT concentration (AST seems particularly high for a control mice). Did the author measure other metabolites that would give a more detailed picture on the liver physiology (i.e albumin or creatinine level or simply the weight of the liver).

3- Using the WB in figure 1 D, can the author show if the IL-1beta is process (running at a lower MW)? Similarly, if the inflammasome is more activated in the Drp1-LiKO, is Gasdermin D activated as well? Also, can the authors confirm in the legend that reason for 2 separate samples in each condition (are they technical or biological duplicate)?

4- Figure 5b: Can the authors confirm that Chop, P8 and Trib3 expression are normalised to one gene in particular? It seems they are all at 1 fold change. If they are not normalised to one gene in

particular, they should be on separated graph.

5- Finally (line 340-341), the authors claim that DRP1 could be targeted for future therapeutic treatments. Could the authors be more precise about this claim, any thoughts how? I guess Drp1 would need to be activated to restore a protective fission. This might prove complicated as for example: Liver-specific ablation of Mfn2 (and thus inducing fission) increases inflammation (Ref13).

Reviewer #3:

Remarks to the Author:

Dr. Wang L et al demonstrated the role of hepatocyte Drp1 in the mechanisms of acute liver injury induced by LPS administration. Authors have prepared hepatocyte-specific Drp1-deficient mice and demonstrated that 1) deletion of hepatocyte Drp1 exacerbated liver injury accompanied with increased cytokine expressions, 2) deletion of Drp1 induced hepatocyte apoptosis accompanied with accumulation of M1 macrophages. Authors also demonstrated impaired mitophagy, induction of ER stress and loss of mitochondrial membrane potential as mechanisms of these phenomenon. Authors are established groups with numerous previous reports using Drp1 conditional knockout mice and used various techniques including histopathological analysis, flow cytometry and state-of-the-art microarray analysis. The presented data support the main conclusion that deletion of Drp1 enhances LPS-induced acute liver injury.

Major comments

1. In the Figure 1, authors have demonstrated time-course of inflammatory cytokines in the serum and liver. These data indicate that inductions of inflammatory cytokines are enhanced in Drp1LiKO mice at 1 hour after LPS injection. In contrast, western blot shows no significant differences in LC3-II at 1 hour after LPS injection (Figure 4a, 4b), suggesting that inductions of these cytokines were enhanced by mechanisms which are independent of mitophagy. Is it explained by ER stress, mitochondrial ROS production or other mechanisms?

2. Authors have demonstrated the deletion of Drp1 induced loss of mitochondrial membrane potential (Figure 5e). Please discuss the mechanisms by which deletion of Drp1 causes loss of mitochondrial membrane potential. Is mitochondrial respiratory chain affected by mitochondrial morphology per se? Or, does Drp1 regulate mitochondrial membrane potential by mechanisms which are independent of mitochondrial morphology, e.g. mPTP (mitochondrial permeability transition pore) opening or MOMP (mitochondrial outer membrane permeabilization)?

3. In the title, it is better to mention about the liver. For example, Drp1 deficiency accelerates LPS-induced acute liver injuries and inflammation, etc. In the current title, it is obscure whether there is a causal relationship between impairment of mitophagy and inflammation. Or, was impairment of mitophagy just observed in association with inflammation?

4. Relative contribution of inflammation in the mechanisms of hepatocyte death in Drp1LiKO mice is obscure. The data of isolated primary hepatocyte suggests that hepatocyte survival is impaired at baseline without inflammation. Macrophage depletion by clodronate liposomes may elucidate the relative contribution of macrophage-mediated inflammation in liver injuries after LPS injection.

Minor comment

1. Page 15, Line 204-206: Please check and correct the sentences.

2. In Page 17, Line 229, authors have described that cell viability of hepatocytes was $100.00 \pm 12.20\%$ in control group. Is it possible that mean + S.E.M. exceeds 100%?

Communications Biology manuscript COMMSBIO-20-2763A (Wang et al)

Point-by- point responses

To Reviewer #1

Reviewers' comments:

Reviewer #1 (Remarks to the Author):

In their study entitled “Dynamin-related 1 protein 1 deficiency impairs mitophagy and accelerates lipopolysaccharide-induced inflammation in mice, Wang et al. investigate the role of DRP1, a key protein in the control of mitochondrial dynamics, in liver inflammation.

This is a very interesting study. Several points need to be improved and/or clarified.

Major points:

1. The authors performed LPS-injection to generate liver dysfunction. It would be interesting to assess the impact of liver DRP1 deficiency in an additional model such as acetaminophen-induced liver dysfunction or in the context of bacterial infection (e.g IP injection of E. Coli or CLP-induced bacterial infection). Indeed, the authors showed an increase in M1 macrophages in the liver. Would it translate into a reduced bacterial load in the live as the inhibition of mitophagy in macrophages was recently shown to induce macrophage activation and to promote anti-bacterial defense (Patoli et al., JCI, Oct 2020)?

<Reply>

Thank you for this comment. We strongly agree that it would be interesting to assess the impact of liver Drp1 deficiency in an additional model such as acetaminophen-induced liver dysfunction or in the context of bacterial infection (especially the cecal ligation and puncture-induced bacterial infection model). However, we are not able to include other models in the current study. We are considering to further address the role of Drp1 in mitophagy formation, macrophage activation and host defense using different liver dysfunction models as well as Drp1^{flox/flox}; Lysm^{Cre/Cre} mice (macrophage *Drp1* specific

knockout mice) in the future.

2. Experimental conditions should be more clearly indicated on the figures. Indeed, it is not always clear whether mice or cells have been treated with LPS. In addition, the authors should systematically assess the same condition i.e WT, WT+LPS, liver KO, Liver KO + LPS. Conditions without LPS exposure are not always depicted.

<Reply>

Thank you for the great suggestion. In the revised manuscript, we have added the group without LPS exposure accordingly and have all groups (control, control+LPS, KO, KO+LPS) systematically assessed (Figs. 3d, 4e, 6a and 6b).

3. In addition to DRP1 mRNA levels in the liver of LPS-treated mice and isolated hepatocytes the authors should also assess DRP1 protein levels (total and phosphorylated forms).

<Reply>

Thank you for this suggestion. The Drp1 protein levels in the liver and isolated hepatocytes, both in total and phosphorylated forms, have been assessed. These new results are included in the revised manuscript as Figs. 2a, 2c, 2d, 6c, and 6d.

4. What is the impact of liver DRP1 deficiency upon LPS treatment in term of mortality?

<Reply>

According to the previous survival study that mice were intraperitoneally injected with 4 different doses of LPS (1.5, 2.5, 5, and 20 mg/kg body weight) and monitored for 7 days, the survival rate of young mice was 100% with LPS dose of 5 mg/kg body weight. In our present study, both control and *Drp1*LiKO mice survived after 24 hours LPS injection (5 mg/kg body weight) as well as 4 hours after LPS/ATP injection (10 mg/kg body weight) at the time of sample collection. As *Drp1*LiKO mice develops more severe symptoms of endotoxin shock, liver dysfunction and increased inflammatory response, DRP1 deficiency may reduce the survival from a high dose of LPS in the long term.

5. Infiltration of macrophages by immune cells should be assessed by flow cytometry and not only by microscopy approaches which are not sufficiently accurate for such a purpose. Why infiltration of the liver by immune cells was not assessed in LPS treated mice? LPS + ATP is a very different model and NLRP3 is also activated upon LPS treatment alone.

<Reply>

Thank you for this comment. LPS or LPS/ATP injection caused liver congestion, such liver damage decreased the efficiency of perfusion and collagenase digestion. Therefore, we were not able to assess the infiltration of the liver by immune cells by flow cytometry.

We agree with you that NLRP3 inflammasome is also activated upon LPS treatment alone as demonstrated in Figs. 2g, 2h, and 2i.

6. The proportion of M1 macrophages should be assessed with classical markers: e.g CD80, CD86, CD64. Ly6C is a marker for monocytes. F4/80⁺-CD11b⁺-Ly6C⁺ cells do not allow to discriminate between monocytes and macrophages. It is surprising that all CD11b-F4/80 positive cells are expressing Ly6C at a similar level. Usually, several populations: Ly6 Low, Ly6C med and Ly6 High can be observed.

<Reply>

Thank you for this suggestion. We assessed the proportion of M1 inflammatory macrophages using two classical makers, CD80 and CD64. We have added these new data to Figs 3e and 3f.

7. How the authors explain the decrease in autophagy in LiDRP1 KO mice treated with LPS?

<Reply>

First, it has been reported that DRP1-mediated mitochondrial fission helps autophagosomes engulf mitochondrial (Youle and Narendra., 2011). We have previously showed by electron microscope analysis that the mitochondria in *Drp1*LiKO mice were swollen and bigger in size (Wang et al., 2015). Our present study showed that mitochondria in *Drp1*LiKO primary hepatocytes were tubular but not fragmented. Thus, the fission defect due to the loss of Drp1 may affect mitochondria autophagosomes engulf leading to impaired autophagy formation. Second, it has been

recently established that autophagosomes form at the mitochondria- ER contact site (MAM) in mammalian cells (Hamasaki et al., 2013). MAM has been suggested as a membrane sub-domain functioning for phospholipid and/ or calcium transport. We have demonstrated that loss of Drp1 led to MAM defect (Wang et al., 2015). In addition, our unpublished mass-spectrometry analysis showed that phosphatidylethanolamine (PE), which conjugates with LC3-I to form LC3-II, is significantly decreased in the liver of *Drp1*LiKO mice. These are the possible mechanisms could lead to the decreased autophagy in *Drp1*LiKO mice.

8. The authors propose that mitophagy is decreased in the hepatocytes of LPS-treated animals. This is consistent with previous studies. Nevertheless, how the authors explain the drop in the mitochondrial membrane potential? Several studies showed that mitochondrial membrane potential is increased when mitophagy or autophagy are inhibited (Zhou R, et al Nature, 2011, 469(7329):221–225; Gomes LC, et al, Nat Cell Biol. 2011;13(5):589–598. Yao Z, et al J Cell Sci. 2011;124(Pt 24):4194–4202, Patoli et al., JCI, Oct 2020).

<Reply>

Thank you for insightful comment. We discussed the mechanisms by which deletion of Drp1 caused the loss of mitochondrial membrane potential on P26-27, lines 362-384 as “Drp1 deletion leads to the loss of $\Delta\Psi_m$ in *Drp1*LiKO mouse primary hepatocytes; since loss of $\Delta\Psi_m$ is closely linked to cell apoptosis by various insults, we believe that the increased population with $\Delta\Psi_m$ collapse in *Drp1*LiKO mouse primary hepatocytes might be an insult and/or a consequence induced by increased cell apoptosis. We have previously reported that, despite the observed morphological changes in mitochondria, the expression levels of the mitochondrial respiratory chain complex and mitochondrial respiratory activity were not changed by loss of DRP1¹⁴. Opening of the mitochondrial permeability transition pore (mPTP) causes inner membrane potential collapse and eventually leads to mitochondrial swelling, rupture, and cell death. In DRP1-deficient hearts and MEFs, inhibition of DRP1-mediated fission produces elongated mitochondria with increased mPTP opening⁵⁰. Furthermore, accelerated mPTP opening in DRP1 null mitochondria has been associated with mitophagy in MEFs and contributes to cardiomyocyte necrosis and dilated cardiomyopathy in mice⁵¹. Mitochondrial outer membrane permeabilization (MOMP) also induces the loss of $\Delta\Psi_m$

and results in the release of apoptotic proteins that activate the downstream pathway of apoptosis. Although the mechanism by which DRP1 participates in MOMP remains to be elucidated, previous studies indicate that DRP1 inhibition has direct effects on MOMP^{52 53} and that the role of DRP1 in MOMP can be distinguished from mitochondrial fission⁵⁴. Several studies have shown that $\Delta\Psi_m$ increases when mitophagy or autophagy are inhibited^{55 56 57}; nevertheless, it is possible to conclude that the increased population observed with $\Delta\Psi_m$ collapse in *Drp1*LiKO mouse primary hepatocytes may have been caused by mPTP opening and/or MOMP, which are independent of mitochondrial fission or mitophagy formation.”

9. The authors claim that DRP1 deficiency promotes inflammation through ER stress (Fig 7). This is not demonstrated in the paper. Such a claim has to be supported by experimental data using KO mice or pharmacological modulation of ER stress in WT and LiDRP1 KO mice or hepatocytes exposed or not to LPS. Is LiDRP1 KO-mediated liver inflammation blunted when P8/Chop/trib3 is inhibited or invalidated? This point is critical as the data do not fully support the model proposed in the discussion.

<Reply>

Thank you for this constructive suggestion. To address your comment, firstly, we examined the ER stress response during LPS treatment in vivo with control and *Drp1*LiKO mice (Figs. 2a, 2e, and 2f). Secondly, DRP1 deficiency promoted inflammation through ER stress was assessed in primary hepatocytes derived from control and *Drp1*LiKO mice. We showed that inflammatory cytokines elevated in *Drp1*LiKO mouse primary hepatocytes during ER stress inducer thapsigargin treatment (Figs. 7a and 7b). As shown below in the “Fig for reviews. 1”, when trib3 was knockdown by SiRNA, expression of inflammatory cytokines (*Tnfa* and *Il6*) was not blunted. In contrast, pharmacological modulation of ER stress using classical ER stress inhibitor TUDCA, blunted *Drp1* defect-mediated inflammation (Fig. 7c). These data support our conclusion that *Drp1* deficiency promotes inflammation through ER stress.

Figure for reviews. 1

Minor:

1. Statistical analysis should be depicted in the figure legends.

<Reply>

Thank you for this comment. Statistical analysis has been added in the figure legends.

2. Fig 1a: Is there a difference in the basal levels of cytokines in WT and liver DRP1 KO mice?

<Reply>

As indicated in Fig. 1a, no significant change in the basal levels of cytokines was observed in the liver of control and *Drp1*LiKO mice according to two-way ANOVA

analysis.

3. Fig 1b: The authors should normalize all mRNA levels to WT mice at t0 and not to the t0 for each genotype.

<Reply>

Thank you for this comment. We did normalize all mRNA levels to control mice at t0. In the revised manuscript, we changed the Y axis to two segments to clearly demonstrate the levels of different groups.

4. Why 7AAD is used for the characterization of M2 macrophages but not for M1 macrophages (since the goal is to exclude dead cells)?

<Reply>

Thank you for this comment. 7AAD has been used to exclude dead cells during re-analysis of the M1 macrophages by flow cytometry.

5. Page 19 line 272 to 278. Explanation regarding mitophagy levels are not clear. This paragraph needs to be clarified as well as the corresponding figure which does not clearly display the experimental conditions.

<Reply>

Sorry for the confusion. In the revised manuscript, condition without LPS exposure has been added and 4 groups (control, control+LPS, KO, KO+LPS) were assessed. The explanation regarding mitophagy levels is also changed accordingly in the text.

To Reviewer #2

Reviewer #2 (Remarks to the Author):

The manuscript by Wang et al. is investigating how the lack of Drp-1 in hepatocytes leads to an increased inflammation. These findings are timely as the role of mitochondrial dynamics in inflammatory diseases is under investigation by numerous research teams. This work uses multiples approaches to confirm their findings and is generally convincing. This is an interesting manuscript and the aims of this paper are clear. There are although some points that would require some clarification:

1- Apoptosis was measured by TUNEL but this type of cell death could be confirmed by other means (caspase 3 activation, cytochrome C activation)? In addition, can the author see M1 macrophages in close proximity to apoptotic hepatocyte in their microscopy?

<Reply>

Thank you for this suggestion. In this revised manuscript, we added results of caspase 3 activation quantified by levels of pro-caspase 3 and cleaved-caspase 3 using Western blot (Figs. 2a and 2b).

For the question on whether M1 macrophages are in close proximity to apoptotic hepatocyte, we are also interested to know. However, our current sections available for microscopic analysis are not conclusive. We plan to address this question in our future experiment by co-immunostaining using M1 macrophage marker such as CD64 and cleaved-caspase 3 antibodies.

2- Furthermore, the mechanism remains unclear on how the lack of Drp1 triggers apoptosis. The authors highlight that Trib3 is more expressed in the Drp1-liKO cells, which could lead to apoptosis. Would inhibition of Trib3 (i.e siRNA) improve hepatocyte survival and decrease apoptosis level (caspase 3)? In the discussion, the authors suggest that ER-stress would increase ROS production, is that the case for those hepatocytes? If so, does reducing ROS increases hepatocytes survival? Ultimately, does treatment with caspase inhibitor (such as z-VAD) could reduce the inflammation level in Drp1-LiKO mice?

<Reply>

1) Thanks for your insightful comment. We conducted the suggested experiment, as shown below in “Figure for reviews. 2”, knockdown of *Trib3* by SiRNA was not able to protect hepatocytes from apoptosis as the cell viability and caspase 3 level remained unchanged. We have modified the discussion accordingly.

Figure for reviews. 2

2) Alternatively, ER stress due to *Drp1* deficiency may increase ROS production and apoptosis. However, as shown in “Figure for reviews 3”, we analyzed the mitochondrial ROS by MitoSOX using flow cytometry and found that the mitochondrial ROS was not increased in *Drp1* defect primary hepatocytes, and the hepatocytes survival was not changed by reducing mitochondrial ROS by the Mito-TEMPO treatment. Therefore, we exclude this possibility and modified the discussion accordingly.

Figure for reviews. 3

- 3) On another note, we observed that Drp1 deletion led to the loss of $\Delta\Psi_m$ in *Drp1*LiKO mouse primary hepatocytes. The loss of $\Delta\Psi_m$ has been closely linked to cell apoptosis. Indeed, *Drp1*LiKO mouse primary hepatocytes exhibit increased population with $\Delta\Psi_m$ collapse. This data supports that deletion of Drp1 acts like an insult to the cells causing the loss of $\Delta\Psi_m$ and cell apoptosis.
- 4) Ultimately, treatment with caspase inhibitor V-ZAD-FMK reduced the inflammation level both in control and *Drp1*LiKO hepatocytes (Fig. 7c).

3- In Figure 5, the number of hepatocytes recovered is dramatically different between the two genotypes. Maybe as a result, it seems that the confluence of the cells is quite different also (by microscopy). How consistent was the cell confluence? Would a difference in the cell confluence have an impact on cellular stress (i.e TMRM)?

<Reply>

Thanks for your insightful comment. The numbers of hepatocytes recovered from primary hepatocytes isolation are dramatically different between control and *Drp1*LiKO mice. We do seed equal number of live-cells to the culture plate after counting cells with vital stain trypan blue. The difference in confluency of control and *Drp1*LiKO cells can be observed after 24 hours post seeding. The cell number was determined using cell count reagent SF (CCR-SF, Nacalai tesque, Japan) which allows sensitive colorimetric assays using utilizing highly water-soluble tetrazolium salt (WST-8) with a microplate reader at the absorbance 450nm. The *Drp1*LiKO cells have a decreased cell viability (60~70% of the control cells). We agree that such a difference in the cell confluence might impact cellular stress as well. As we discussed in the manuscript, the increased population with $\Delta\Psi_m$ collapse in *Drp1*LiKO mouse primary hepatocytes may be caused by mPTP opening and/ or MOMP.

4- It might be worth discussing the unclear function of Drp1 in inflammation upon the cell type. The authors show that hepatocytes lacking Drp1 have an increased inflammatory response but several recent publications (PMID: 32686869, PMID: 30637805, PMID: 23815397) have also shown that macrophages lacking drp-1 have a lower inflammatory response. Are macrophages and hepatocyte physiology different in regard to Drp1-dependent apoptosis?

<Reply>

We are grateful that the reviewer reminded us to look into this important point. A short answer is yes, the cell contexts could affect the outcomes of Drp1 deletion. We discussed the physiology different of macrophage and hepatocyte in regard to Drp1-dependent apoptosis (P25, lines 352-357) “Our study reveals that hepatocytes lacking DRP1 have an increased inflammatory response, whereas several recent studies have shown that a lack of DRP1 leads to lower inflammatory responses in macrophages and microglia^{17 18}. Hepatocytes and macrophages might be physiologically different in terms of DRP1-dependent apoptosis. Therefore, it will be worth studying and clarifying the function of DRP1 in apoptosis and inflammation according to cell type.”

Minors comments:

1- Mitofusin and not Mitofusion (Line 43)

<Reply>

Sorry for the typographical error. We have corrected accordingly.

2- As there is an important difference in apoptosis (re the number of primary hepatocytes harvested), one can wonder if the liver of the Drp1-LiKO are physiologically perturbed. Indeed, in Fig 1c, one can see a small but constantly increases AST and ALT concentration (AST seems particularly high for a control mice). Did the author measure other metabolites that would give a more detailed picture on the liver physiology (i.e albumin or creatinine level or simply the weight of the liver).

<Reply>

Thank you for this comment. In our previous study (Wang et al., 2015), we have reported that on the normal chow diet, no apparent difference was found in the liver weight of the *Drp1*LiKO mice compared with controls. We also measured other metabolites which gave a more detailed picture on the liver physiology (Shown in the Table below).

No.	TP	ALB	BUN	CRE	Na	K	Cl	Ca	IP	LDH	AMY	r-GT
-----	----	-----	-----	-----	----	---	----	----	----	-----	-----	------

	(g/dL)	(g/dL)	(mg/dL)	(mg/dL)	(mEq/L)	(mEq/L)	(mEq/L)	(mg/dL)	(mg/dL)	(IU/L)	(IU/L)	(IU/L)
Drp1WT 1	5.8	3.3	24.2	0.10	156	6.2	118	8.7	5.3	382	2423	3>
Drp1WT 2	5.4	3.0	38.0	0.11	157	6.7	121	8.6	4.9	441	1959	3>
Drp1WT 3	4.9	3.0	26.1	0.11	152	6.6	111	8.5	6.5	515	1927	3>
Drp1KO 1	5.0	2.8	32.1	0.10	155	7.3	118	8.2	5.9	838	1846	3>
Drp1KO 2	5.4	3.1	40.4	0.12	154	6.8	117	8.4	4.8	584	1784	3>
Drp1KO 3	5.0	3.0	35.9	0.11	154	6.7	116	8.4	5.5	586	1999	3>

3- Using the WB in figure 1 D, can the author show if the IL-1beta is process (running at a lower MW)? Similarly, if the inflammasome is more activated in the Drp1-LiKO, is Gasdermin D activated as well? Also, can the authors confirm in the legend that reason for 2 separate samples in each condition (are they technical or biological duplicate?)?

<Reply>

Thank you for this comment. It is well known that IL-1beta is produced as a 31 kDa pro-peptide and is converted into active 17 kDa form by caspase 1. Running at a gradient gel (5-20%), the 17 kDa product was not detected. The serum and liver Gasdermine D levels were determined by Gasdermine D ELISA Kit and significant increases in serum gasdermin D levels at 8 h and 24 h after LPS injection were observed in *Drp1LiKO* comparing to control mice (Fig. 2i).

We confirmed in the legend that reason for 2 separate samples in each condition. The following sentence has been added in the figure legends (Figs. 2a, 2g, 4a, 4f, and 6c), “The biological duplicate samples in each condition numbered as #1 and #2.”

4- Figure 5b: Can the authors confirm that Chop, P8 and Trib3 expression are

normalised to one gene in particular? It seems they are all at 1 fold change. If they are not normalised to one gene in particular, they should be on separated graph.

<Reply>

Thank you for this comment and sorry for mistake. We have plotted separated graphs showing in Figure. 7a. The fold changes of *Chop*, *P8* and *Trib3* mRNA expression (PBS treated group) between the control and *Drp1*LiKO cells remained similar to the previous data.

5- Finally (line 340-341), the authors claim that DRP1 could be targeted for future therapeutic treatments. Could the authors be more precise about this claim, any thoughts how? I guess Drp1 would need to be activated to restore a protective fission. This might prove complicated as for example: Liver-specific ablation of Mfn2 (and thus inducing fission) increases inflammation (Ref13).

<Reply>

Thank you for this important suggestion. We agree that Drp1 would need to be activated to restore a protective fission. We have added the following statement to discussion (P 28-29, Lines 404-409) “Under acute inflammatory liver disease, DRP1 could be potentially activated to restore a protective fission in hepatocytes. In recent years, lipid nanoparticles have been developed to passively and actively target drugs to the liver. Liver-targeted lipid nanoparticles with the expression of DRP1 or ablation of MFN2 by siRNA treatment (thereby inducing fission) could be used to produce anti-inflammatory activities.”

To Reviewer #3

Reviewer #3 (Remarks to the Author):

Dr. Wang L et al demonstrated the role of hepatocyte Drp1 in the mechanisms of acute liver injury induced by LPS administration. Authors have prepared hepatocyte-specific Drp1-deficient mice and demonstrated that 1) deletion of hepatocyte Drp1 exacerbated liver injury accompanied with increased cytokine expressions, 2) deletion of Drp1 induced hepatocyte apoptosis accompanied with accumulation of M1 macrophages. Authors also demonstrated impaired mitophagy, induction of ER stress and loss of mitochondrial membrane potential as mechanisms of these phenomenon. Authors are established groups with numerous previous reports using Drp1 conditional knockout mice and used various techniques including histopathological analysis, flow cytometry and state-of-the-art microarray analysis. The presented data support the main conclusion that deletion of Drp1 enhances LPS-induced acute liver injury.

Major comments

1. In the Figure 1, authors have demonstrated time-course of inflammatory cytokines in the serum and liver. These data indicate that inductions of inflammatory cytokines are enhanced in Drp1LiKO mice at 1 hour after LPS injection. In contrast, western blot shows no significant differences in LC3-II at 1 hour after LPS injection (Figure 4a, 4b), suggesting that inductions of these cytokines were enhanced by mechanisms which are independent of mitophagy. Is it explained by ER stress, mitochondrial ROS production or other mechanisms?

<Reply>

Thanks for your insightful and constructive suggestion. We have examined the ER stress response during LPS treatment using *Drp1*LiKO mice (Figs. 2a, 2e, and 2f). We showed that inflammatory cytokines elevated in *Drp1*LiKO mouse primary hepatocytes during ER stress inducer thapsigargin treatment (Figs. 7a and 7b). Importantly, pharmacological modulation of ER stress using classic ER stress inhibitor TUDCA blunted Drp1 defect-mediated inflammation (Fig. 7c). These data supported our conclusion that Drp1 deficiency promotes inflammation through ER stress.

Previously, we also suggested that ER stress would increase ROS production. However, as shown in “Figure for reviews 3”, the mitochondrial ROS levels measured by MitoSOX using flow cytometry didn’t increase in DRP1 defect primary hepatocytes. The hepatocytes’ survival was not changed by reduction of mitochondrial ROS using Mito-TEMPO. Therefore, our final conclusion is: DRP1 deficiency promotes inflammation through ER stress rather than mitochondrial ROS.

Figure for reviews. 3

2. Authors have demonstrated the deletion of Drp1 induced loss of mitochondrial membrane potential (Figure 5e). Please discuss the mechanisms by which deletion of Drp1 causes loss of mitochondrial membrane potential. Is mitochondrial respiratory chain affected by mitochondrial morphology per se? Or, does Drp1 regulate mitochondrial membrane potential by mechanisms which are independent of mitochondrial morphology, e.g. mPTP (mitochondrial permeability transition pore) opening or MOMP (mitochondrial outer membrane permeabilization)?

<Reply>

Thank you for this comment. We have added discussion of the mechanisms with the following statement (P26-27, lines 362-384) “DRP1 deletion leads to the loss of $\Delta\Psi_m$ in *Drp1LiKO* mouse primary hepatocytes; since loss of $\Delta\Psi_m$ is closely linked to cell apoptosis by various insults, we believe that the increased population with $\Delta\Psi_m$ collapse in *Drp1LiKO* mouse primary hepatocytes might be an insult and/or a consequence induced by increased cell apoptosis. We have previously reported that, despite the

observed morphological changes in mitochondria, the expression levels of the mitochondrial respiratory chain complex and mitochondrial respiratory activity were not changed by loss of DRP1¹⁴. Opening of the mitochondrial permeability transition pore (mPTP) causes inner membrane potential collapse and eventually leads to mitochondrial swelling, rupture, and cell death. In DRP1-deficient hearts and MEFs, inhibition of DRP1-mediated fission produces elongated mitochondria with increased mPTP opening⁵⁰. Furthermore, accelerated mPTP opening in DRP1 null mitochondria has been associated with mitophagy in MEFs and contributes to cardiomyocyte necrosis and dilated cardiomyopathy in mice⁵¹. Mitochondrial outer membrane permeabilization (MOMP) also induces the loss of $\Delta\Psi_m$ and results in the release of apoptotic proteins that activate the downstream pathway of apoptosis. Although the mechanism by which DRP1 participates in MOMP remains to be elucidated, previous studies indicate that DRP1 inhibition has direct effects on MOMP^{52 53} and that the role of DRP1 in MOMP can be distinguished from mitochondrial fission⁵⁴. Several studies have shown that $\Delta\Psi_m$ increases when mitophagy or autophagy are inhibited^{55 56 57}; nevertheless, it is possible to conclude that the increased population observed with $\Delta\Psi_m$ collapse in *Drp1*LiKO mouse primary hepatocytes may have been caused by mPTP opening and/or MOMP, which are independent of mitochondrial fission or mitophagy formation.”

3. In the title, it is better to mention about the liver. For example, Drp1 deficiency accelerates LPS-induced acute liver injuries and inflammation, etc. In the current title, it is obscure whether there is a causal relationship between impairment of mitophagy and inflammation. Or, was impairment of mitophagy just observed in association with inflammation?

<Reply>

Thank you for this important suggestion. Now the title of this revised manuscript is “Dynamin-related protein 1 deficiency accelerates lipopolysaccharide-induced acute liver injury and inflammation in mice”

4. Relative contribution of inflammation in the mechanisms of hepatocyte death in Drp1LiKO mice is obscure. The data of isolated primary hepatocyte suggests that hepatocyte survival is impaired at baseline without inflammation. Macrophage

depletion by clodronate liposomes may elucidate the relative contribution of macrophage-mediated inflammation in liver injuries after LPS injection.

<Reply>

Thank you for this comment. We agree that hepatocyte survival is impaired without inflammation. This might be caused by the increased population with $\Delta\Psi_m$ collapse in *Drp1*LiKO cells that may be caused by mPTP opening and/ or MOMP. We appreciate your comment that macrophage depletion by clodronate liposomes may elucidate the relative contribution of macrophage-mediated inflammation in liver injuries after LPS injection. We would like to try this new method in our future study.

Minor comment

1. Page 15, Line 204-206: Please check and correct the sentences.

<Reply>

Thank you for this comment. These sentences have been corrected as the following:
“Selective autophagy of mitochondria, known as mitophagy, is an important mitochondrial quality control mechanism³⁷. Phosphatase and tensin homolog-induced putative kinase 1 (PINK1) is an established mediator of mitophagy^{38 39 40}.”

2. In Page 17, Line 229, authors have described that cell viability of hepatocytes was 100.00±12.20% in control group. Is it possible that mean + S.E.M. exceeds 100%?

<Reply>

Thank you for this comment. We are sorry for this mistake. The cell viability of hepatocytes was shown in the absolute number of the live-cells not in percentage.

Reviewers' Comments:

Reviewer #1:

Remarks to the Author:

The authors addressed most of the raised questions. I am satisfied with the provided answers and the additional data. The work is suitable for publication. This is an interesting study that provides new insights regarding the role of mitochondrial dynamics in the hepatocytes and in liver injuries.

Reviewer #2:

Remarks to the Author:

The manuscript by Wang et al. is investigating how the lack of Drp-1 in hepatocytes leads to an increased inflammation. It has been modified and improved following reviewer's comments.

Wang et al have looked into my comments about the link between DRP1 and cell death, mitoROS and other clarification in the text and discussion. They have also now extended their discussion about membrane potential of the mitochondria. Thus I have no other comments on the manuscript.

Last note is that, unless I missed it, it would have been more convenient to have a Track-Change version of the manuscript as it was quite hard to pinpoint where the authors have made their changes.

Reviewer #3:

Remarks to the Author:

The authors responded politely to questions and concerns from reviewers, including the addition of experimental data. It is considered that the content has improved compared to the first draft and the scientific value of this paper has also increased.

Communications Biology manuscript COMMSBIO-20-2763A (Wang et al)
Point-by- point responses

Reviewers' comments:

Reviewer #1 (Remarks to the Author):

The authors addressed most of the raised questions. I am satisfied with the provided answers and the additional data. The work is suitable for publication. This is an interesting study that provide new insights regarding the role of mitochondrial dynamics in the hepatocytes and in liver injuries.

<Reply>

We greatly appreciate your thoughtful comments and suggestions that helped improve the manuscript. We appreciate your time and efforts on improving this manuscript.

Reviewer #2 (Remarks to the Author):

The manuscript by Wang et al. is investigating how the lack of Drp-1 in hepatocytes leads to an increased inflammation. It has been modified and improved following reviewer's comments.

Wang et al have looked into my comments about the link between DRP1 and cell death, mitoROS and other clarification in the text and discussion. They have also now extends their discussion about membrane potential of the mitochondria. Thus I have no other comments on the manuscript.

Last note is that, unless I missed it, it would have been more convenient to have a Track-Change version of the manuscript as it was quite hard to pinpoint where the authors have made their changes.

<Reply>

We feel sorry that we did not provide a Track-Change version of the manuscript. We would like to apologize for this carelessness. We sincerely appreciate your time and efforts on improving this manuscript.

Reviewer #3 (Remarks to the Author):

The authors responded politely to questions and concerns from reviewers, including the addition of experimental data. It is considered that the content has improved compared to the first draft and the scientific value of this paper has also increased.

<Reply>

We sincerely appreciate all valuable comments and suggestions, which helped us to improve the quality of the article.